# Towards improving short-term sea ice predictability using deformation observations

Anton Korosov[1], Pierre Rampal[2,1], Yue Ying[1], Einar Ólason[1], and Timothy Williams[1]

[1]Nansen Environmental and Remote Sensing Center, Jahnebakken 3, Bergen, 5007, Norway
[2]CNRS, Institut de Géophysique de l'Environnement, Grenoble, France

**Correspondence:** Anton Korosov (anton.korosov@nersc.no)

**Abstract.**

Short-term sea ice predictability is challenging despite recent advancements in sea ice modelling and new observations of sea ice deformation, that capture small-scale features (open leads and ridges) at kilometre scale. A new method for assimilation of satellite-derived sea ice deformation into numerical sea ice models is presented. Ice deformation provided by the Copernicus Marine Service is computed from sea ice drift derived from Synthetic Aperture Radar at a high spatio-temporal resolution. We show that high values of ice deformation can be interpreted as reduced ice concentration or increased ice damage - scalar variables responsible for ice strength in brittle or viscous-plastic sea ice dynamical models. This method is tested as a proof-of-concept with the neXt-generation Sea Ice Model (neXtSIM), where the assimilation scheme uses a data insertion approach and forecasting with one member. We obtain statistics of assimilation impact over a long test period with many realisations starting from different initial times. Assimilation and forecasting experiments are run on synthetic and real observations in January 2021 and show increased accuracy of deformation prediction for the first 3 – 4 days. Similar conclusions are obtained using both brittle and viscous-plastic rheologies implemented in neXtSIM. Thus, the forecasts improve due to the update of sea ice mechanical properties, rather than the exact rheological formulation.

It is demonstrated that the assimilated information can be extrapolated in space — gaps in spatially discontinuous satellite observations of deformation are filled with a realistic pattern of ice cracks, confirmed by later satellite observations. Limitations and usefulness of the proposed assimilation approach are discussed in a context of ensemble forecasts. Pathways to estimate intrinsic predictability of sea ice deformation are proposed.

## 1 Introduction

Sea ice in the Arctic is continuously drifting and deforming under the influence of atmospheric winds and ocean currents (Sverdrup, 1950; Colony and Thorndike, 1984; Rampal et al., 2009). In summer, when ice concentration is low and ice extent is small, the sea ice is mostly in free drift — the speed and direction of the drift are dominated by the atmospheric and ocean drag forces and by the Coriolis force. In contrast, in winter the sea ice covers almost the entire Arctic ocean and its adjacent seas, forming a rigid and nearly continuous solid plate. As a consequence, sea ice does not drift freely anymore, but instead exhibits an intermittent drift with localised deformation. First, under increasing external forcing the undamaged ice deforms

primarily as an elastic material. Internal stresses gradually accumulate in the material until a failure criterion is reached, which corresponds to a limit when sea ice fractures, and then the ice starts deforming along the multiple narrow and elongated cracks, and does so until these later refreeze or when the load (winds and currents) on the ice changes. Location, density and orientation of these cracks greatly control the overall and individual motion of the resulting ice pieces, from small floes (∼10m) to large plates (∼100km).

Under divergent ice motion these cracks become open leads, significantly increasing ocean-air heat and mass exchange and modifying local atmospheric boundary layer and ocean mixed layer (Olason et al., 2021). Open leads are also key both for marine fauna survival, and for facilitating ship navigation. Under convergent or shear motions sea ice ridges are formed along the cracks. Ridge sails and keels affect the drag by winds and currents. At the same time ridged ice significantly impedes navigation in the Arctic (Lindsay and Stern, 2003).

Given the importance of sea-ice fracturing for air–sea–ice interface processes, marine life and navigation, its accurate monitoring and forecasting is in great demand. Observations of cracks can be performed using satellite remote sensing by retrieval of high-resolution sea ice drift from Synthetic Aperture Radar (SAR) data and computation of sea ice deformation components (Kwok et al., 1990). The Radarsat Geophysical Processor System (RGPS) dataset was the first attempt to systematically observe sea ice drift and derive sea ice deformation on high spatial resolution (10 km) and with high frequency (3 days) over a

long period of time (the winters from 1996—2016) (Kwok, 1998). An operational SAR-based sea ice drift and deformation product is currently provided by the Copernicus Marine Services (Saldo, 2020). It is derived from Sentinel-1 C-band SAR data at 12-hour / 10-km resolution. The cracks appear on satellite-derived ice deformation products as narrow (10 - 30 km, depending on resolution of satellite data) and long (up to 1000 km) lineaments and are also called linear kinematic features (LKFs) (Kwok, 2001).

To address the challenge of realistic simulation and forecasting of sea ice dynamics the next-generation sea ice model (neXtSIM, Bouillon and Rampal, 2015; Rampal et al., 2016) was developed based on elasto-brittle sea ice rheology (Girard et al., 2011). The spatio-temporal scaling properties of ice deformation are simulated correctly by neXtSIM (Rampal et al., 2019) and the distribution of cracks looks very realistic (Olason et al., 2022). In a recent model intercomparison paper (Bouchat et al., 2022) neXtSIM results ranked among the best for simulating the observed probability distribution, spatial distribution and

fractal properties of sea ice deformation. Analysis of spatial and temporal scaling (Fig. 13 in Bouchat et al. (2022)) shows that the spatial structure function of neXtSIM matches the RGPS observations very well, whereas the temporal one is overestimated by 3 – 5 %, probably indicating some overestimation of the intermittency by neXtSIM. In the aforementioned studies neXtSIM was run with a similar setup: the dynamic equations were solved on at triangular mesh with 10 km resolution using finite element method.

Despite the recent advances in the sea ice modeling, the exact timing and spatial distribution (including orientation, width, length and angle of fracture) of strong deformation zones, or LKFs, is not yet predicted precisely. Moreover, there are many sources of uncertainty in LKF forecasting that require additional research including: uncertainties in atmospheric and ocean forcing; rheology and model parametrisation; model numerics; initial conditions for sea ice states; observing network and

data assimilation. Mohammadi-Aragh et al. (2018) evaluated the potential predictability of LKFs using an ensemble of sea ice models all using a viscous-plastic rheology, but the practical predictability remains unknown.

The primary goal of our research is, therefore, to improve the skill in predicting LKFs by assimilating novel satellite observations of sea ice deformation. Our secondary goal is to quantify the practical predictability of LKFs by the neXtSIM model when combined with satellite observations via data assimilation (DA), and study factors affecting it.

Several methodological and technical challenges with assimilating sea ice deformation into a model are worth mentioning here. First, the "direct insertion" method operates in the model state space. However, the observed deformation is not a model prognostic variable, so an operator is used to convert deformation to the model variables. This operator is an inverse of the observation operators used in data assimilation, since it maps from the observation space back to the state space. There is also no guarantee that updated model variables will remain accurate during a forecast. For example, the ice drift is a model variable, but it is strongly dependent on external forcing and increments from assimilation will only survive a short period of time. Second, the observed cracks are very localized in space and time, which pose challenges in modelling its covariance structure for data assimilation methods such as 3DVar (Lorenc, 1986). Ensemble Kalman filter (EnKF) (Evensen, 2003) is potentially a good solution through estimating a flow-dependent covariance structure from an ensemble of model runs. However, the current ensemble data assimilation framework for neXtSIM (Cheng et al., 2020) is not ready to assimilate deformation yet. Therefore, and also as a proof of concept, we present here a first attempt to assimilate sea ice deformation into neXtSIM using a simple direct data insertion scheme (Stanev and Schulz-Stellenfleth, 2014), and perform a sensitivity analysis useful for demonstration of the approach.

The concept of sea ice deformation assimilation is presented in Section 2, followed by a detailed description of satellite observations of deformation and the methodology for assimilation and running forecasting experiments in Section 3. The results are presented and discussed in Sections 4 and 5.

## 2   Link between observed ice deformation and model state

The central idea in our assimilation approach is that the ice in the model should become weaker — in a mechanical sense — where high deformation is observed. In the current context, we simulate sea ice "weakness" evolution according to the Brittle Bingham-Maxwell (BBM) rheology (see (Olason et al., 2022) for details on how this rheology is implemented into neXtSIM). BBM belongs to a family of brittle rheologies, with earlier variations being the Elasto-Brittle (EB) (Girard et al., 2011) and the Maxwell Elasto-Brittle (MEB) (Dansereau et al., 2016). Two regimes are distinguished in the BBM: the undamaged pack ice can have small elastic (reversible) deformations; in the cracks the deformation is visco-elastic (partly permanent and partly reversible) and can become quite high (e.g. several percent per day over a spatial scale of about 10 km). The BBM stress evolution equation writes as follows:

$$\dot{\sigma} = E\mathbf{K} : \dot{\varepsilon} - \frac{\sigma}{\lambda}\left(1 + \widetilde{P} + \frac{\lambda\dot{d}}{1-d}\right), \tag{1}$$

where $\sigma$ is the internal stress tensor, $E$ is the ice elasticity, $K : \dot{\varepsilon}$ is the stiffness tensor, $\lambda = \eta/E$ is the viscous relaxation time, $\widetilde{P}$ is a generalised friction term, $d$ is the ice damage (with $d = 0$ being completely undamaged ice).

Elasticity and viscosity are functions of the model state variables damage (d) and sea ice concentration (A):

$$E = E_0(1 - d)e^{-C(1-A)} \tag{2}$$

$$\eta = \eta_0(1 - d)^{\alpha}e^{-\alpha C(1-A)}, \tag{3}$$

where $E_0$ and $\eta_0$ are the undamaged elasticity and viscosity, and $\alpha > 1$ is a constant. It should be noted that there are two ice categories in the model: young ice, which is formed during water freezing, and older ice which is formed after young ice exceeds a threshold in thickness (Rampal et al., 2019). Only the older ice concentration (referred to as $A$) is used in the rheological equations.

$\widetilde{P}$ contains the effects of the friction element and is defined as:

$$\widetilde{P} = \begin{cases} \frac{P_{\max}}{\sigma_n} & \text{for } \sigma_n < -P_{\max}, \\ -1 & \text{for } -P_{\max} < \sigma_n < 0, \\ 0 & \text{for } \sigma_n > 0. \end{cases} \tag{4}$$

The friction element is active when damaged ice is converging (i.e., when the normal stress $\sigma_n$<0); when $\sigma_n < -P_{\max}$, the frictional forces can no longer balance the convergence and the ice starts to ridge. This threshold is defined as:

$$P_{\max} = Ph^{3/2}e^{-C(1-A)}, \tag{5}$$

where $P$ is a constant scaling parameter for the ridging threshold to parameterise $P_{\max}$, following the results of Hopkins (1998), and $h$ is thickness.

Eqs. 2, 3, 5 show that increasing damage ($d$) will decrease viscosity, while decreasing concentration $A$ will both decrease viscosity and shift the threshold $P_{\max}$, so that the ice transitions from the elastic to the viscous regime and allows larger deformations without significant increase in internal stress.

We use an empirical function to convert the observed deformation to model variables, so that the update can take place in the model state space. The "observed" model variables damage $d_o$ and concentration of older ice $A_o$ are derived from the observed deformation $\varepsilon_o$ using the following experimental formulations:

$$d_o = H'_d(\varepsilon_o) \tag{6}$$

$$A_o = H'_A(\varepsilon_o) \tag{7}$$

where $H'_d$ and $H'_A$ are inverse observational operators (see Section 3.3 and Appendix 1).

## 3 Data and Methods

### 3.1 Satellite observations of sea ice deformation

We used the sea ice drift and deformation dataset from Copernicus Marine Services (Saldo, 2020) acquired in January 2021. The dataset comprises gridded products derived from Sentinel-1 synthetic aperture radar (SAR) images, with 10 km spatial resolution. Ice drift is computed from pairs of images separated by approximately 24 hours and the product is delivered every 12 hours. The spatial coverage of the product is irregular - the East Siberian, Laptev and Kara seas and the polar gap (north of 87°N) are never covered, while other Arctic regions are observed at least once nearly every 24 hours.

### 3.2 Simulation experiments setup

The model is run using a 900-s time step on a triangular mesh with 10 km spatial resolution covering the Arctic ocean and adjacent seas north of 65°N. The model is forced with the latest version (Cycle 45r1) of the Integrated Forecast System European Center for Medium-Range Weather Forecasts (ECMWF) (Owens and Hewson, 2018) and the TOPAZ4 (Sakov et al., 2012) ocean forcing fields (currents, sea surface temperature, sea surface salinity).

Experiments start from 1 December 2020 ($t_0$) and last for two months. Let $\mathbf{x}$ denote the model state variables (e.g., concentration, damage, drift, etc.), $\mathbf{x}_{t_0}$ is the initial condition, and $\mathcal{M}_{t_n \to t_{n+1}}$ is the non-linear model (neXtSIM) to propagate state from time $t_n$ to $t_{n+1}$.

Let $\mathbf{y}$ denote the observations (ice deformation rate), which is related to the model state variables through $\mathbf{y}_t = H(\mathbf{x}_t)$, where $H$ is the observation operator. Real satellite observations $\mathbf{y}_t^o$ are available throughout the test period. Although the deformation rate is derived from sea ice drift, derived from Radarsat-2 SAR images, we call them "observations of deformation" as opposed to "simulation of deformation" by neXtSIM.

In the first experiment a verifying "truth run" is generated:

$$\mathbf{x}_t^{\mathrm{tr}} = \mathcal{M}_{t_0 \to t}(\mathbf{x}_{t_0}) \tag{8}$$

The period before 1 January 2021 is used as a spin-up time, the data from $\mathbf{x}_t^{\mathrm{tr}}$ is not used, and time $t_1$ denotes the 1 January 2021. Then four sets of 10-days forecasts are initialised and ran every day in January 2021, so that each set has 31 forecast (see scheme on Fig. 1).

**1. Forecasts initiated from truth**:

$$\mathbf{x}_{t_1 \to t}^{\mathrm{T}} = \mathcal{M}_{t_1 \to t}(\mathbf{x}_{t_1}^{\mathrm{tr}}) + \psi_t, \tag{9}$$

where $\psi_t$ is a random noise added to the model operator due to uncertainties in model numerics that cause the forecasts run to differ from the truth run. These forecasts are evaluated by computing the error in observation space:

$$\epsilon_{\delta t}^T = \left\langle H(\mathbf{x}_{t \to t+\delta t}^T) - H(\mathbf{x}_{t \to t+\delta t}^{tr}) \right\rangle \tag{10}$$

where $\langle \cdot \rangle$ denotes averaging over the different forecasts starting from $t_1, t_2, \ldots, t_n$, i.e. $\left\langle \epsilon_{t \to t+\delta t}^T \right\rangle = \sum_{t=t_1}^{t_n} \epsilon_{t \to t+\delta t}^T$, then plotted w.r.t. lead time $\delta t$.

**2. Forecasts without data assimilation:**

$$\mathbf{x}_{t_1 \to t} = \mathcal{M}_{t_1 \to t}(\mathbf{x}_{t_1}) + \psi_t \tag{11}$$

The first forecast is initiated from $t_0$, subsequent forecasts are initiated from the outputs of the previous forecasts. The forecasts initiated during the spin-up period are not used. The forecasts after 1 January are evaluated against truth:

$$\epsilon_t^B = H(\mathbf{x}_t) - H(\mathbf{x}_t^{tr}) \tag{12}$$

and against real observations:

$$\epsilon_t^O = H(\mathbf{x}_t) - \mathbf{y}_t^o \tag{13}$$

During the spin-up period $\epsilon_t^B$ grows and reaches its saturation level $\epsilon_B$, which we consider to be the climatological level for this error. Since the forecasts without data assimilation don't see real data, the error $\epsilon_t^O$ averaged over one month ($\epsilon_O$) can also
be considered as the climatological level.

**3. Forecasts with assimilation of synthetic data:**

$$\mathbf{x}_{t_i \to t}^{as} = \mathcal{M}_{t_i \to t}(\mathbf{x}_{t_i}^{as}) + \psi_t \tag{14}$$

where $i$ denotes days in January 2021 (e.g., January $1^{st}$, January $2^{nd}$, January $3^{rd}$, etc), $\mathbf{x}_{t_i}^{as}$ is the analysis of synthetic observations from the truth run and the forecasts without assimilation performed at $t_i$: $\mathbf{x}_{t_i}^{as} = \mathcal{A}(\mathbf{x}_{t_i}, \mathbf{y}_{t_i}^{tr}; H', \mathbf{w})$. In the assimilation
scheme in this paper, we use the inverse operator $H'$ to compute model state (concentration and damage) from the observed deformation, $\mathbf{x}_t = H'(\mathbf{y}_t)$ (see Eqs. 6 and 7 for how $H'$ is constructed) and $\mathbf{w}$ are the tuning parameters (see Eqs. 22 for how $\mathcal{A}$ is constructed). These forecasts are evaluated with:

$$\epsilon_{\delta t}^S = \left\langle H(\mathbf{x}_{t \to t+\delta t}^{as}) - \mathbf{y}_{t+\delta t}^{tr} \right\rangle \tag{15}$$

**4. Forecasts with assimilation of real satellite data:**

$$\mathbf{x}_{t_i}^{ar} = \mathcal{A}(\mathbf{x}_{t_i}, \mathbf{y}_{t_i}^o; H', \mathbf{w})$$

$$\mathbf{x}_{t_i \to t}^{ar} = \mathcal{M}_{t_i \to t}(\mathbf{x}_{t_i}^{ar}) + \psi_t \tag{16}$$

evaluated with:

$$\epsilon_{\delta t}^A = \left\langle H(\mathbf{x}_{t \to t+\delta t}^{ar}) - \mathbf{y}_{t+\delta t}^o \right\rangle \tag{17}$$

It should be noted that for assimilation the deformation is computed from observations of drift at $t_{n-1} \to t_n$ and the model is initialised from the analysis at time $t_n$. Then the forecast is compared with deformation computed at $t_n \to t_{n+1}$ (corresponding to lead time $\delta t = 1$), $t_{n+1} \to t_{n+2}$ ($\delta t = 2$), etc. Thus, the error of the forecast $\epsilon_{t_n \to t_{n+1}}^A$ is independent from observations used
in assimilation $\mathbf{y}_{t_{n-1} \to t_n}^o$ (the same holds for $\epsilon_{\delta t}^S$).

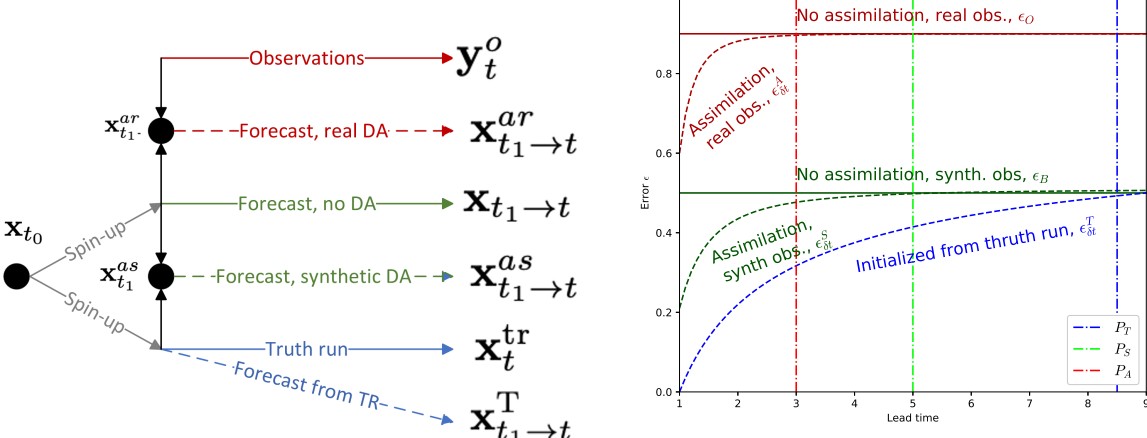

**Figure 1.** Scheme of experiments (left) and scheme of errors (right). The truth run is shown by solid blue line, observations - by solid red line. The forecasts initiated from the truth - dashed blue line; without assimilation - solid green line; with assimilation of synthetic observations - dashed green line; with assimilation of real satellite observations - dashed red line. Grey lines show spin-up period for the truth and the no-DA forecasts. On the scheme of errors the lines are coloured as follows: red - evaluation against satellite observations, green - evaluation against synthetic observations, blue - evaluation against the "truth run". Solid lines show climatological level, dashed - average over forecasts. Vertical lines $P_T$, $P_S$ and $P_A$ indicate potential predictability, practical predictability for synthetic DA and practical predictability for real DA, correspondingly.

### Definition of predictability

Predictability is defined as the time at which a forecast error reaches a background level (Zhang et al., 2019). Since the errors of the forecasts without assimilation $\epsilon_B$ and $\epsilon_O$ are at their respective saturation levels, we assume them to be the background levels for the forecasts with assimilation. Therefore, in a perfect-model scenario (forecast with initialisation from truth) the intrinsic predictability is the time $\delta t$ when $\epsilon_{\delta t}^T \approx \epsilon_B$. The practical predictability is the time $\delta t$ when $\epsilon_{\delta t}^S \approx \epsilon_B$. Similarly, in the case of assimilation of real observations the practical predictability is time $\delta t$ when $\epsilon_{\delta t}^A \approx \epsilon_O$.

Our previous experiments (Williams et al., 2021) showed that assimilation of concentration is not significantly affecting accuracy of sea ice drift forecast. In that sense the reference forecasts without DA ($\mathbf{x}_{t_1 \to t}$) are almost equal to the forecasts with assimilation of concentration, and the error $\epsilon_A$ helps to evaluate the general improvement due to the DA system.

### 3.3 Inverse observational operator

The inverse observational operator $H'$ is a function to compute a model state variable from observations: $\mathbf{x} = H'(\mathbf{y})$. Since reliable simultaneous observations of concentration and deformation at scales of 1 day / 10 km are not available, and damage is not an observable variable, we cannot use an empirical inverse operator. I.e., $H' \neq H'_o$, where $H'$ is the inverse operator in

question, and $H'_o$ is the empirical inverse operator between the observed total sea ice concentration ($x_o$) and sea ice deformation ($y^o$): $x^o = H'_o(y^o)$. Ultimately, the purpose of the inverse operator is not to describe a physically realistic process (i.e., linear decrease of concentration due to divergence) but to minimise the error of the deformation forecast: $E^f_n = y^o_n - y_n = y^o_n - H \circ \mathcal{M} \circ \mathcal{A} \circ H'(y^o_{n-1})$, where $H$ is the forward operator to compute deformation from ice drift, $\mathcal{M}$ is the numerical model to forecast ice drift, $\mathcal{A}$ is the assimilation of the inverse operator applied to the observed total deformation at the previous time step.

In our experiments, the deformation is computed in each model mesh element by integrating ice drift velocities simulated in the truth run over a period of 24 hours ($t_{n-1} \to t_n$):

$$\mathbf{y}_{t_n} = \varepsilon_{t_n} = H(\mathbf{x}_{t_{n-1} \to t_n}) \tag{18}$$

Then $H'$ is applied to $\mathbf{y}_{t_n}$ for computing damage and concentration, and the results are compared to the simulated damage and concentration in the corresponding elements at the end of this period ($t_n$). The initial values of $H'$ parameters are found by minimisation:

$$H' = \underset{H'}{\mathrm{argmin}} \left( \sum_{n=1}^{30} \left[ H'(\mathbf{y}_{t_{n-1} \to t_n}) - \mathbf{x}_{t_n} \right]^2 \right) \tag{19}$$

where $n$ denotes day number in the truth run.

The total deformation ($\varepsilon_{tot}$, m$^{-1}$) is used as a predictor for damage and concentration under the assumption that all deformation events (convergence, divergence and shear) indicate the presence of weaker ice that may continue to be deformed. Ice weakness is simulated in neXtSIM by decreased concentration or increased damage (see Eqs. 2 and 3). Observation of any deformation components (including convergence) is interpreted in the assimilation procedure as an increase in ice weakness and, therefore, a decrease in concentration or an increase in damage. Since $A$ and $d$ are the components of sea ice strength, and the total deformation is a good proxy for the presence of weak ice, we suggest building the inverse operator $H'$ on the assumption that $A$ and $d$ are related to the total deformation. It should be emphasised, that only the older ice fraction $A$ is updated in the assimilation procedure and the total ice concentration remains the same.

The inverse operators for damage and concentration have the following form (for further details see Appendix 1):

$$H'_d(\varepsilon_{tot}) = 1 - 10^{k_2 + k_3 \log_{10}(\varepsilon_{tot})} - k_1 \tag{20}$$

$$H'_A(\varepsilon_{tot}) = 1 - a_1 \varepsilon_{tot} \tag{21}$$

## 3.4 Data assimilation method

We update the damage and concentration variables in the model according to the observed deformation using a simple "direct data insertion" approach as a proof of concept for DA (Stanev and Schulz-Stellenfleth, 2014). The updated state variable is computed as a weighted average of the forecasted variable ($\mathbf{x}$) and the variable computed from the observed deformation ($\mathbf{x}^o = H'(\mathbf{y}^o)$):

$$\mathbf{x}^a = w\mathbf{x}^o + (1 - w)\mathbf{x} \tag{22}$$

where $w$ is the weight applied to observations.

As defined here, the weight can be interpreted as the precision (the inverse of the uncertainty) of the observed relative to the modelled variable. In variational assimilation schemes the uncertainties are characterised by error co-variance matrices, while here we assume no correlation structure between variables and only characterise the relative precision (signal to noise ratio of observation-to-model variable error variances) as a single weight. However, we still allow this weight to be individually specified for different variables and also to be spatially varying, giving some more flexibility to the update scheme. Also, we note that we assume the model variables are spatially uncorrelated, and that the variable on each model mesh point can be updated independently. Since sea ice deformation is accommodated along nearly 1D geometrical features (i.e. fractures), correlation can only usually be seen along the fracture, and so the assumption of low spatial correlations in all other directions is reasonable.

We parameterize the weights as:

$$w = w_v W \tag{23}$$

where $w_v$ is a variable specific weight (either $w_d$ or $w_A$) and $W$ is a weight that is dependent on observed deformation:

$$W = \begin{cases} 1, & \text{if } \varepsilon > \varepsilon_{min}, \\ 0, & \text{otherwise} \end{cases} \tag{24}$$

where $\varepsilon_{min}$ is a threshold for total deformation found in sensitivity experiments. It is known that low values of deformation have higher uncertainty (Dierking et al., 2020), so it is sensible to update model variables only when the observed deformation exceeds the threshold value. This threshold localizes the impact of assimilating observed deformation to only be effective in the vicinity of ice cracks.

The variable-dependency is tested by setting the weight $w_d$ or $w_a$ to 0, i.e., letting the assimilation to update either damage, or concentration, or both to see the impact of the update.

## 3.5 Sensitivity experiments

The list of parameters tested in the sensitivity experiments is provided in Table 1. The values for $a_1$ and $\varepsilon_{min}$ were tested within reasonable ranges, i.e. expected decrease of concentration due to ice deformation given the observable ranges of deformation.

**Table 1.** Tested parameters of sea ice deformation assimilation scheme

| Parameter | Description | Eq. | Tested values |
|---|---|---|---|
| $a_1$ | Coefficient for computing sea ice concentration from deformation, [% d] | 21 | 0.1, 0.5, 0.9, 1.2, 1.5, 2 |
| $w_d$ | Weight of damage assimilation | 23 | 0, 0.5, 1 |
| $w_A$ | Weight of concentration assimilation | 23 | 0, 0.5, 1 |
| $\varepsilon_{min}$ | Threshold of total deformation for applying assimilation, [d$^{-1}$] | 24 | 0.02, 0.1 |

Over 64 experiments were run following the algorithm:

– Choose assimilation parameters from a predefined space and save in a configuration file

      – Run a series of 31 forecasts in January 2021 with these parameters

      – Evaluate each forecast by comparing simulated and observed deformations

      – Average the evaluated quality metrics over the month

All combinations of parameter values (except for ineffective ones, e.g., $w_c = 0$ and $w_d = 0$) were tested.

The effect of assimilation on the prediction skill is evaluated by comparison of the simulated and observed total deformation fields as it is crucial information for safe navigation. The forecasts are evaluated using two quality metrics: area of maximum cross-correlation (hereafter referred to as $A_{MCC}$), and difference in probability distributions of total deformation ($KS$).

     $A_{MCC}$ is computed as the area where the maximum cross-correlation (MCC, see Appendix 2 for details) is above 0.35, normalised by the total area of available satellite observations. $A_{MCC}$ indicates the level of spatial collocation of forecast

LKFs to observations at a relatively fine spatial scale (1 - 2 pixels, 10 - 20 km). Unlike the LKF evaluation metrics suggested in (Hutter et al., 2019), that compare only statistical properties of LKFs (number, density, length, orientation, etc), the MCC-based metric estimates co-alignment of individual LKFs on model simulations and satellite observations. It is also thought to be more sensitive to LKFs with low deformation magnitude, as no threshold is applied for their detection.

     $KS$ is the difference between cumulative distribution functions of $\varepsilon_{tot}$ computed using the Kolmogorov-Smirnov test

(Smirnov, 1939), and indicates the correspondence of the magnitude of the predicted deformation to observations on pan-Arctic scale.

## 4    Results

### 4.1    Impact on fields of concentration and damage

Fig. 2 shows example fields of sea ice deformation computed from ice drift between 15 and 16 January 2021 and also the

260 corresponding damage and concentration fields computed during the assimilation procedure using the following values: $\varepsilon_{min} =$ 0.02, $w_d = 1$, $w_c = 1$, $a_1 = 0.9$ . White gaps on the field of deformation (Fig. 2, A) show areas without satellite data coverage. The white gaps on the reconstructed concentration and damage maps are also due to application of the $\varepsilon_{min}$ threshold - values of $\varepsilon_{tot}$ below that threshold are not used in assimilation.

     The range of the reconstructed concentration corresponds well to the simulated one. Only the largest cracks with deformation

above $0.3\,\mathrm{d}^{-1}$ have concentration below 0.7, while the other cracks have a realistic values of concentrations in range 0.9 - 1 if compared, for example, to the AMSR2 sea ice concentration product from the Ocean and Sea Ice Satellite Application Facility (EUMETSAT, 2021). The pattern of LKFs, exhibited as reduced concentrations, in general looks similar to the simulated field, but the exact position is, of course, different. It also seems that there are more reconstructed LKFs than the simulated ones. It can be explained by the fact that the simulated concentration only decreases in the case of divergence, whereas the

reconstructed LKFs are a function of total deformation.

Similar conclusions can be drawn regarding the damage fields. The only observed difference is that the simulated damage is so spatially heterogeneous that contributions from the "observed" damage are difficult to spot on the analysis field.

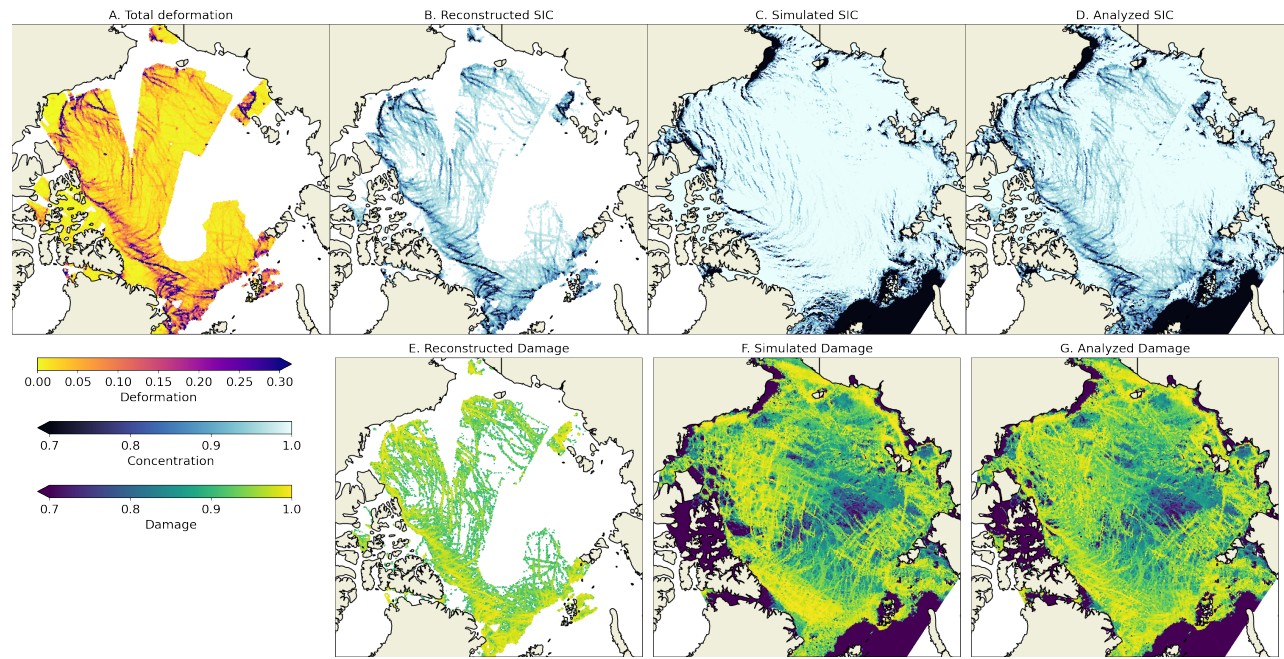

**Figure 2.** A: sea ice deformation $(d^{-1})$ computed from observed sea ice drift between 15 and 16 January 2021. B and E: sea ice concentration and damage reconstructed from the observed deformation using Eqs. 20 and 21. C and F: concentration and damaged simulated by neXtSIM on 16 January. D and G are the analysis (Eq. 22) without further integration.

## 4.2 Impact of assimilation on deformation fields

The impact of assimilation is demonstrated by a comparison of deformation fields from a 3-day forecast without DA (Fig. 3, first column), from observations (Fig. 3, second column), and a forecast with DA (Fig. 3, third column). The assimilation was performed on 16 January (see Fig. 2 for corresponding fields of damage and concentration). The fourth column shows the MCC computed between the observation and the forecast with DA, where insignificant correlations are masked with white colour. In the fifth column the increase of MCC ($MCC_{Increase} = MCC_{DA} - MCC_{NODA}$) is shown as an indication of areas where DA improved location of the LKFs.

Fig. 3 clearly shows that the field of deformation predicted without DA is different from the observations both in terms of location, sharpness and orientation of cracks, as well as in terms of the deformation magnitude. The first day after assimilation the field of predicted deformation is substantially different from the no-DA run. Visually the position of some cracks correspond well to the observations, which is supported by high values of correlation (the average correlation is 0.7 — see Fig. 3, fourth column) and MCC is higher in the DA-forecast in most of the observed areas (the Beaufort, Chuckhi, Lincoln seas). On the second day, while both forecasts start to look more similar, the MCC with observations is still high in many areas, but an

increase of MCC is visible only in parts of the Beaufort sea. On the third day the improvement which is likely introduced by DA is obvious only in the central Arctic, on intersection of two large cracks crossing the entire ocean.

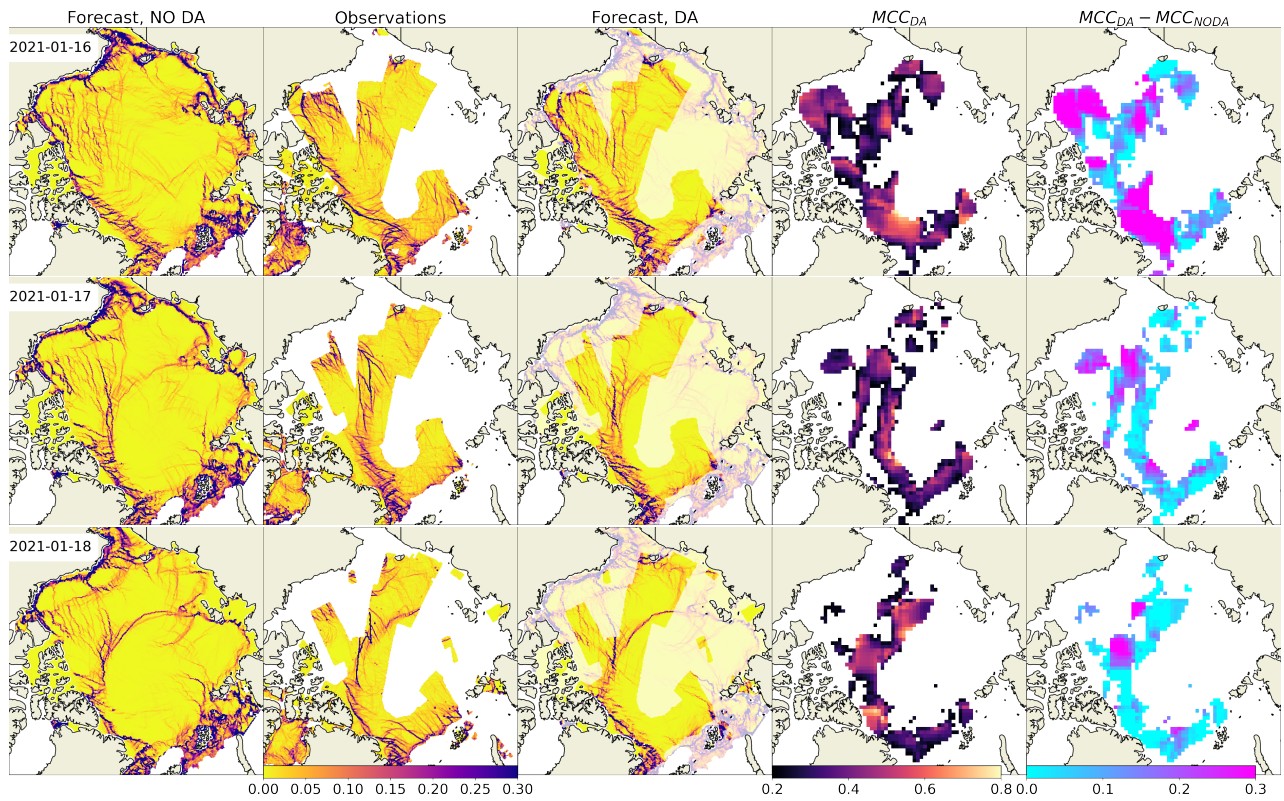

**Figure 3.** Maps of sea ice total deformation ($d^{-1}$) and their comparison. Column 1: forecast without DA; Column 2: observations; Column 3: forecast with DA; Column 4: MCC between observations and forecast with DA; Column 5: increase of MCC due to DA.

The impact of the assimilation on the areas outside of the satellite data coverage can be illustrated on two examples with assimilation of real (Fig. 4) and synthetic (Fig. 5) data. These observations were assimilated in a limited area (indicated by grey colour on the respective figures) on 22 January and the forecast was compared to observations (area-limited satellite observations or pan-Arctic synthetic observations) on 23 January. Visual comparison of forecasts and observations, as well as the maps of MCC increase show that the correlation has improved not only in the area covered by the assimilated observations but also outside it, although, to a lesser degree.

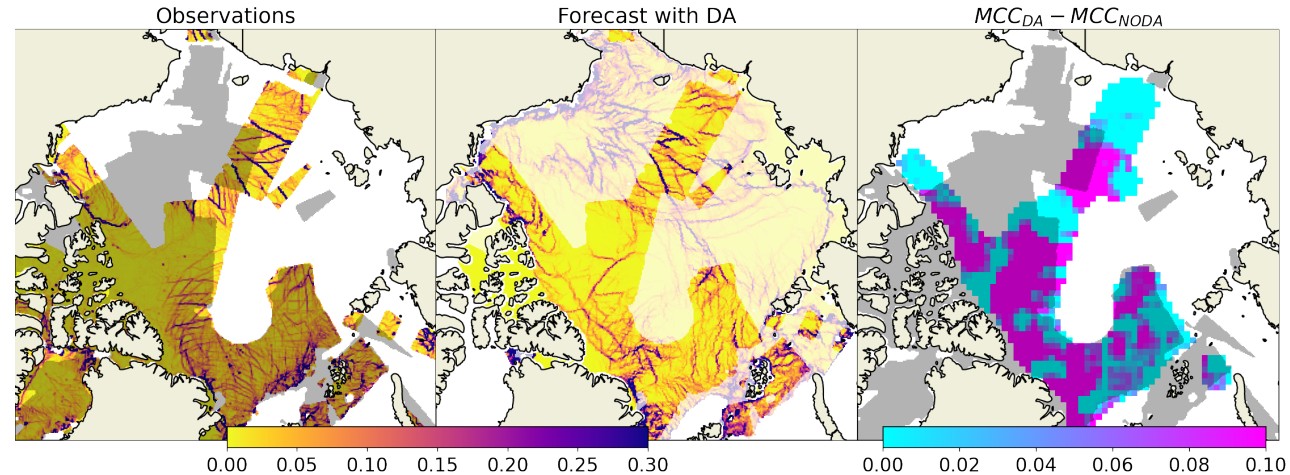

**Figure 4.** Maps of observed (left panel) and simulated (centre panel) deformation on 23 January 2021 and increase of maximum cross-correlation due to DA (right panel). The grey area on the left and right panels shows the extent of data assimilated on 22 January 2021.

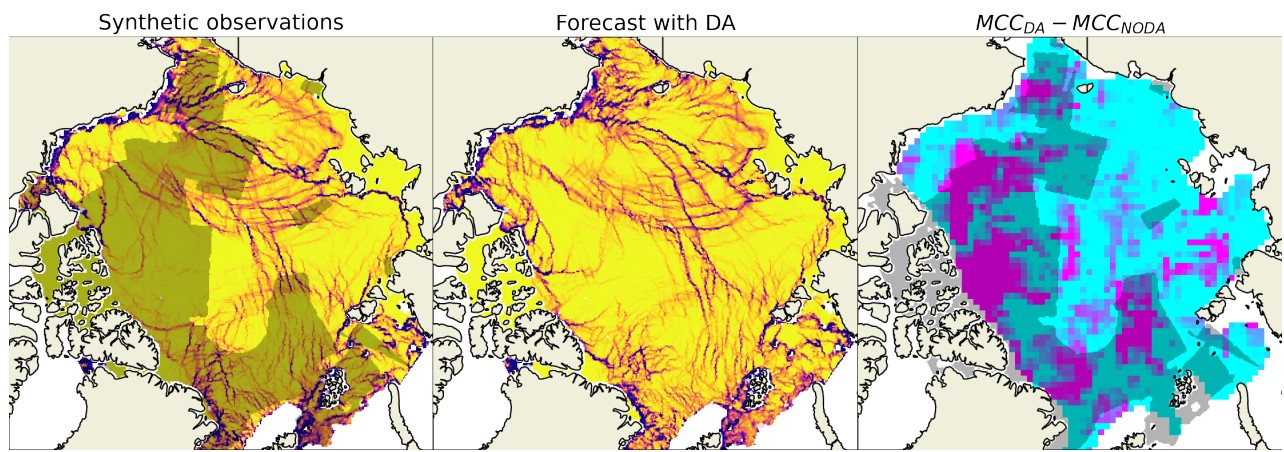

**Figure 5.** Same as Fig. 4 but for synthetic observations.

### 4.3 Practical predictability

The errors $\epsilon_{\delta t}^T$, $\epsilon_B$, $\epsilon_O$, $\epsilon_{\delta t}^S$ and $\epsilon_{\delta t}^A$ (see Eqs. 10, 12, 13, 15, 17, correspondingly) were computed for each forecast and averaged over 31 forecasts. The errors with lead time (shown on Fig. 6) evolve as expected (as in Fig. 1) — the forecast initiated from the truth run has the lowest error, which grows slower than those from the other forecasts. The error $\epsilon_{\delta t}^T$ does not reach the background level $\epsilon_B$, and we can conclude that the intrinsic predictability is larger than 10 days. In forecasts with assimilation of synthetic data the forecast error is initially larger and reaches the background on the $4^{th}$ day, whereas in forecasts with assimilation of satellite observations the error has already reached the background level by the $3^{rd}$ day. Thus, we can say that practical predictability is 4 and 3 days when assimilating synthetic and real observations (respectively).

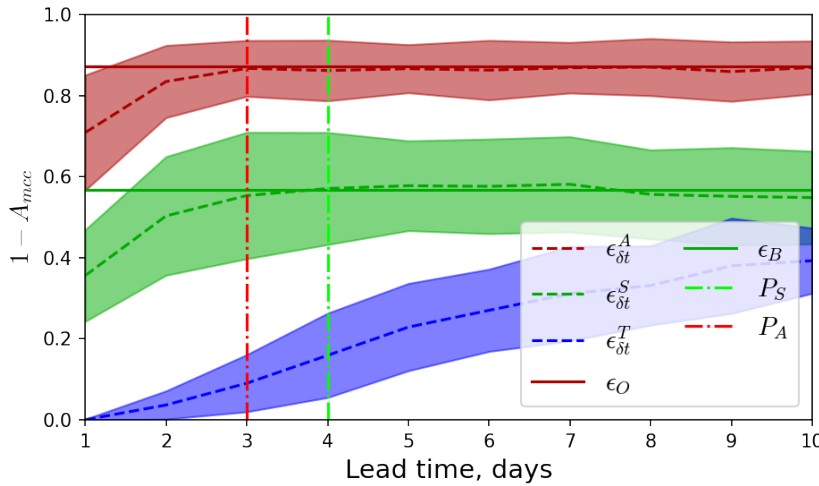

**Figure 6.** Evolution of errors of the forecasts. Line styles and colouring correspond to Fig. 1, the filled area shows one standard deviation.

### 4.4 Sensitivity to assimilation parameters

The results of the sensitivity experiments are summarised in Fig. 7, where the dependence of forecast error (presented as $1 - A_{MCC}$ and $KS$) on values of the assimilation parameters is presented for the first three days of the forecasts. The results

show that the forecast error is very sensitive to the $a_1$ and $w_c$ parameters, is somewhat sensitive to the $\varepsilon_{min}$ parameter, but has almost no sensitivity to the $w_d$ parameter. In other words, assimilation of damage has little impact on forecast error, whereas assimilation of concentration plays a big role.

With $a_1 = 0$ or $w_c = 0$ the $1 - A_{MCC}$ error is the highest and increasing $a_1$ or $w_c$ leads to decrease of this error, indicating that the stronger the inserted reduction of concentration, the large the correlation between forecasts and observations. However,

if the concentration is modified too much during the assimilation ($a_1 > 1$ and $w_c > 0.5$) the forecast deformation increases in magnitude too much and the $KS$ error also starts to grow fast. The forecasts with lead times of 1 day are most impacted, but similar dependencies are also visible in forecasts with lead times of 2 and 3 days.

Increasing the $\varepsilon_{min}$ parameter leads to a slow increase of both the $1 - A_{MCC}$ and $KS$ errors, particularly on the first day of forecast. It can be concluded that even very spatially localized assimilation, when $\varepsilon_{min} = 0.1$, quite considerably impacts the

fields of deformation: the quality is only slightly lower than in the forecasts with $\varepsilon_{min} = 0.02$.

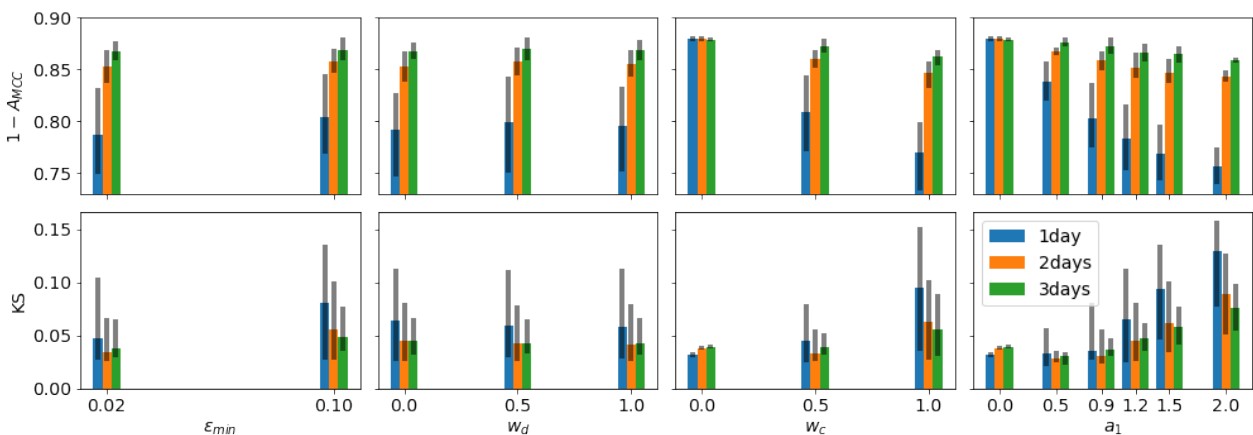

**Figure 7.** Dependence of the errors ($1 - A_{MCC}$ and $KS$) on the assimilation parameters. The coloured bars show averages over all experiments and vertical black lines show 1 standard deviation. Colours denote different lead times.

For selecting the best parameters we plot their values against the quality metrics $1 - A_{MCC}$ and $KS$ on Fig. 8 for each individual experiment. These scatter-plots show:

- that $1 - A_{MCC}$ is inversely proportional to $KS$, i.e., with higher correlation the difference in the probability distributions of the total deformation is also larger.

- $\varepsilon_{min} = 0.02$ provides better results almost in all experiments.

- $w_d$ has no impact on the metrics.

- $w_c = 1$ provides the best results when $a_1$ is 0.9 or 1.2.

- Decrease in $w_c$ can be somewhat compensated by increase in $a_1$, but the results are still worse than with $w_c = 1$.

Based on these results, the best parameter choice for our configuration appears to be:: $\varepsilon_{min} = 0.02$, $w_d = 1$, $w_c = 1$, $a_1 = 0.9$.

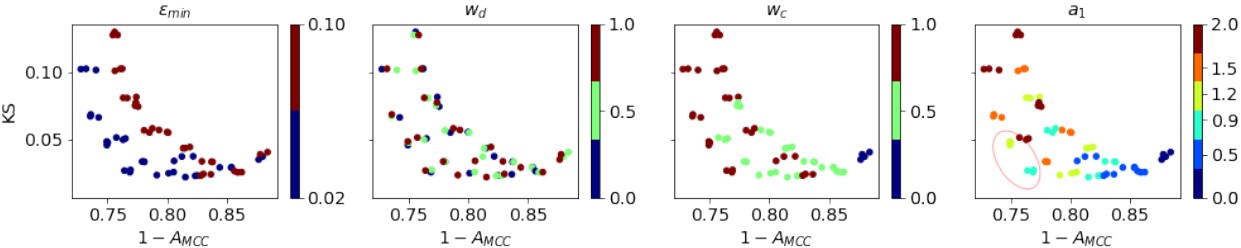

**Figure 8.** Results of individual sensitivity experiments as scatter-plots of $1 - A_{MCC}$ error on X-axis and $KS$ error on Y-axis with discrete values of parameters shown by colour (with a corresponding discrete colorbar). The red circle on panel $a_1$ shows the points with the lowest $1 - A_{MCC}$ and $KS$ metrics that were selected as the best parameters.

## 5 Discussions

### 5.1 Theoretical and practical usefulness

We present a proof-of-concept of assimilation of satellite-derived sea ice deformation which increases the accuracy of deformation prediction for the first $3-4$ days. The approach we used to update the model fields is relatively simple — the concentration

and/or damage are computed from the observed sea ice deformation and inserted into the simulated fields using weighted averaging. Our study demonstrates in practice that information contained in the observed deformation fields can be used for initialisation of model state variables, and shows the time scales over which the forecast of deformation can be improved. Our experiments illustrate that even if data insertion is spatially limited by satellite observations (or even very localized in high deformation zones) it can realistically extrapolate the deformation pattern by connecting the elements of linear kinematic

features, in accord with results from a simple uniaxial loading experiment using viscous-plastic rheology (Ringeisen et al., 2019). Finally, the relative importance of the assimilation parameters (e.g., $a_1$ vs. $\varepsilon_{min}$) and, as explained below, the relative importance of the model state variables are revealed.

The experiments, in which we minimised the difference between simulation and observations by tuning the parameters in a grid search, can be interpreted as an optimisation of the DA hyperparameters. These parameters can be associated with

340 uncertainties in observed deformation, which are either spatially constant ($a_1$, $w_c$ and $w_d$), or spatially varying ($\varepsilon_{min}$). These uncertainties can be related to the diagonal terms in the error covariance matrices used in more sophisticated EnKF and 4DVar methods. However, the uncertainty of the model concentration and damage is either not known or not taken into account. Further study is needed to derive a full covariance matrix, especially the off-diagonal terms depicting cross-variable relations. Knowledge of this obvious weakness in the presented approach paves the road for the planning of future experiments: an

345 ensemble of neXtSIM runs (with perturbed forcing) should be used for evaluating uncertainties in the model variables; detection of covariance between the observed deformation and the model state (not just damage and concentration); and eventually updating the model state using state-of-the-art DA techniques.

## 5.2 Impact of damage and concentration assimilation

As indicated in Olason et al. (2022), neXtSIM is a damage propagation sea ice model and damage is used for changing elasticity and viscosity. So why can't we see the impact of damage assimilation in our experiments? We believe there are two major reasons for that. First, both inverse operators $H'_d$ and $H'_A$ assume linear dependence on the observed total deformation (see Eqs. 20, 21), however stiffness reduces linearly with damage and exponentially with concentration (see Eqs. 2, 3). Therefore, for a similar increment in the observed deformation, the impact on stiffness is larger through concentration, than through damage.

The second reason is that the damage is acting in the model at much smaller timescales than our observations of sea ice deformation. Damage can increase from 0 to 1 in just a few model steps before it eventually starts to decay due to a thermodynamical healing mechanism. The increase of damage takes only a few minutes of simulated time, during which apparent sea ice elasticity and viscosity are proportionally decreased and large-scale and permanent deformation is allowed, accompanied by sea ice internal stresses relaxation. The available observations of deformation are taken on time scales of 24 hours and cannot detect such rapid processes.

The hypothesis that concentration and damage act on different time scales was tested in an idealised twin-experiment: an initially intact ice field ($d = 0$ and $A = 1$ everywhere) was 'broken up' along realistic LKFs. In one experiment, the elements in the LKFs were initiated by reducing concentration to 0.65 and in another one - by increasing damage to 1. The evolution of damage and concentration in several thousand elements with broken-up and intact ice was studied (see Fig. 9).

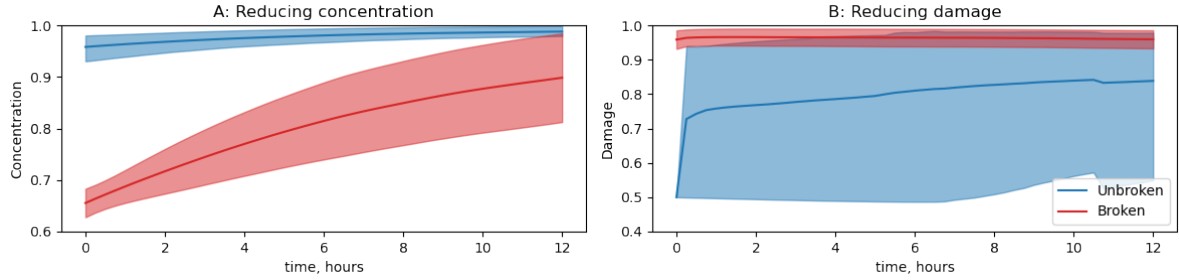

**Figure 9.** Mean and standard deviation of concentration (A) and damage (B) in experiment with breaking ice by reducing concentration (A) and reducing damage (B). Red lines show average for broken elements, and blue lines – unbroken (background).

The study shows that in case when LKFs are initiated by reduced concentration (Fig. 9, A) the situation is quite simple: concentration of ice in the unbroken elements is stably high, and in the broken elements it is first low and then stably increasing due to freezing (and convergence).

For the second experiment (Fig. 9, B) the situation is quite different: in the initially unbroken elements the average damage remains relatively low (0.7 - 0.85), but damage variations are very large with standard deviation reaching 0.2. This happens because in some unbroken elements, that surround the initiated cracks, the internal stress exceeds the Mohr-Coulomb envelope and damage increases up to 1 on very short time scales (few time steps). Further, a cascade of damage events occurs in the

neighbour elements of these newly broken elements. Probability of break-up (damage increase) is higher in directions of high internal stress. Thus, the information about the initiated damage is almost instantly forgotten — it is masked by many newly damaged elements.

Large scale observations of deformation at hourly frequency could probably be used to test our hypothesis of how damage propagates in reality, and illustrate whether or not assimilation of damage indeed leads to a more accurate deformation field **on small time scales**. However, we assimilate and validate against daily observations that show only long term memory in ice weakness expressed in reduced ice concentration.

## 5.3   Experiments with mEVP rheology

To test the feasibility of improving sea ice deformation forecasts with viscous-plastic models, two additional experiments were run with the modified Elasto-Visco-Plastic rheology (mEVP, Bouillon et al., 2013) that has been implemented in neXtSIM (Olason et al., 2022). In these experiments we reduced the fraction of older ice ($A$), as this is the variable affecting the sea ice strength ($P = P^* h e^{-C(1-A)}$, Hibler, 1979). In one experiment $a_1$ (Eq. 21) was set to the 0.9 and in the other one to 2, with $\varepsilon_{min} = 0.02$ and $w_c = 1$ in both experiments. The model with the mEVP rheology was run with the default parameters as
described by Olason et al. (2022).

Comparing Figures 3 and 10, one can notice that the mEVP-based forecasts without assimilation are smoother and the magnitude of total deformation is lower, which agrees well with the finding of Olason et al. (2022). On the first day after assimilation (Fig. 10, upper row) sharp features appear in the places where the LKFs were assimilated. Some of the features correspond well to the observations which is reflected in high values on the $MCC_{DA}$ map. The area where the correlation with
observation in the forecast with DA is better than without DA (pink colour on the $MCC_{DA} - MCC_{NODA}$ map) is similar to the area in the BBM based experiments (Fig. 3). On the second and third days after assimilation (second and third rows on Fig. 10) the impact of assimilation becomes weaker: the sharp features gradually become smoother and disappear, and both the correlation and the area with DA impact decrease.

Looking at the average growth of forecast error (Fig. 11 compares experiments with assimilation or real observations using
the BBM and mEVP rheologies), we can see that the error is nearly the same on the first day, but it grows faster for the mEVP experiments, reaches a higher value ($\approx 0.9$) and saturates later. With $a_1 = 2$ the error grows somewhat slower, but reaches a similar saturation level. We conclude that the assimilation of satellite-derived deformation through the reduction of concentration improves LKF forecasts in both brittle and viscous-plastic models. With a similar setup (i.e. spatial resolution equals 10 km, Lagrangian advection scheme, Finite Element integration method) the model based on a viscous-plastic rheology
produces forecasts with a higher error.

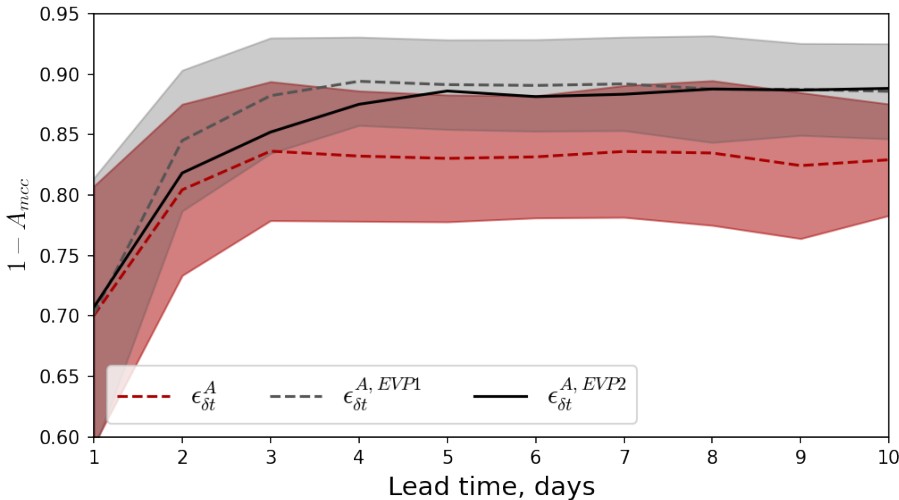

**Figure 10.** Same as Fig. 3 but with mEVP rheology experiment ($a_1 = 0.9$).

**Figure 11.** Evolution of errors of forecasts based on the BBM (red lines) and mEVP (gray and black lines) rheologies. EVP1 denotes the experiment with $a_1 = 0.9$, EVP2: $a_2 = 2$. Other notations correspond to Fig. 6.

## 5.4 Towards evaluation of short-term sea ice predictability

How predictable sea ice features are at kilometre and daily scales still remains an open question. Mohammadi-Aragh et al. (2018) gives a first estimate of the potential predictability of LKFs to be 4–8 days using MITgcm ensemble runs perturbed with atmospheric conditions from the ECMWF Ensemble Prediction System. They also found that additional perturbations in the initial sea ice thickness do not contribute significantly to the forecast error growth in LKFs. The current study provides a new predictability estimate in a different context. Our results show that the deterministic forecast of LKFs gains prediction skill for 3 – 4 days after assimilating deformation observations, indicating a clear impact of improving accuracy of sea ice initial conditions.

More generally, in real world applications, the prediction skill of sea ice LKFs depends on several sources of uncertainties in the system (listed below), so further studies are needed to address each of them and to build a complete picture of the current prediction skill of sea ice at daily time scales and the room for future improvements.

- **Uncertainties in atmospheric and ocean forcing.** The accuracy of contemporary atmospheric and ocean forecasts is quite high (Zhang et al., 2019; Xie et al., 2017). Nevertheless, while forcing the ice model with slightly inaccurate wind fields or ocean currents may only slightly change the ice drift pattern, the ice deformation, being a spatial derivative, will be affected more. Surface wind variability is an important source of sea ice uncertainties (Rabatel et al., 2018; Cheng et al., 2020). A recent study showed that increasing the accuracy (resolution) of atmospheric boundary condition improves the representation of extreme sea ice breakup events in the neXtSIM during the passage of polar cyclones (Rheinlænder et al., 2022). More comprehensive studies are needed to evaluate the impact of external forcing uncertainties on sea ice LKF forecasts at daily time scales.

- **Rheology and model parameterization.** Uncertainties in rheological parameters were shown to be another error source for sea ice forecasts (Urrego-Blanco et al., 2016; Cheng et al., 2020). The BBM rheology (Olason et al., 2022) was implemented in neXtSIM quite recently to replace the previous Maxwell Elasto-Brittle rheology of Dansereau et al. (2016). It was only calibrated by comparing statistical properties of sea ice deformation derived from the RGPS observation dataset (Kwok et al., 1990), and large-scale sea ice thickness and drift time series, and has not yet been tuned for predicting the exact position of cracks in the sea ice cover, which may impact the predictability we obtain in this study. Tuning of the BBM rheology regarding the speed of damage propagation, modifying the constitutive relation, or improving the numerical implementation of the co-evolution of stress, damage and concentration could improve the practical predictability of LKFs. The mEVP rheology could also be further tuned by changing the number of sub-cycles or adding a damage parameter (e.g., as in Savard and Tremblay, 2023).

- **Model numerics.** In neXtSIM, the model equations are derived and solved on a triangular mesh that deforms with the ice motion in a pure-Lagrangian approach. In addition to the physics of the rheological model, this Lagrangian approach may contribute to improving the localisation of cracks in space and time. However, in this framework a remeshing procedure is used when the mesh becomes too distorted in order to replace too skewed triangles with nearly isosceles triangles.

After the remeshing procedure, the model variables are interpolated from the old to the new mesh using a conservative interpolation via supermesh construction. This results in a diffusion of the model fields and likely impacts the predictive skill of the model. Ongoing work of implementing the BBM rheology in an Eulerian version of the neXtSIM model, using a Discontinuous Galerkin advection scheme, will allow us to study the impact of the use of a fixed Eulerian grid compared to a Lagrangian mesh on the efficiency of the data assimilation method and sea ice deformation predictability.

– **Initial conditions for sea ice states.** The impact from uncertainties in initial conditions can be revisited using neXtSIM with the new BBM rheology. Future studies could run ensembles of neXtSIM simulations with perturbation of ice thickness (mean or distribution), concentration, damage, and ice type variables to assess propagation of errors between variables and across scales and to evaluate their impact on predictability.

– **Observation network and data assimilation** In practice, the choice of DA method and availability of observations will also impact the accuracy of initial conditions and therefore impact the predictability. In this study, we made a first attempt to assimilate deformation derived from the operationally available sea ice drift product from Copernicus Marine Services, which provides information at daily time scales for sea ice features. Future studies can assess how observations on different scales (e.g. with higher spatial and temporal resolution) impact the predictability. Also, DA performance can be further improved in future studies using more sophisticated methods to further improve the accuracy in initial conditions.

# 6 Conclusions

The presented method for assimilation of satellite-derived sea ice deformation into a sea ice model efficiently inserts information about where the ice is mechanically weak and improves forecasts of ice deformation for the first $3 - 4$ days. Despite using a relatively simple data insertion approach, the spatially discontinuous satellite observations of deformation are extrapolated by the model, connecting the elements of linear kinematic features in a realistic manner. The main idea behind the proposed method is to relate local sea ice weakness to local reduced ice concentration and increased ice damage, which are computed as functions of observed ice deformation. Notably, this approach was tested with both BBM and mEVP rheologies implemented in neXtSIM, therefore our results can be generalised to both viscous-plastic and brittle frameworks that are used by a wide community of sea ice modellers. The experiments with the parameters of the DA scheme show that updating concentration substantially improves predictive skills on the synoptic scale, whereas updating damage has an effect only on time scales of a few hours, which is difficult to confirm by satellite observations. It is anticipated that update of the ice damage with more frequent observations will play a bigger role in increasing the accuracy of the short range forecasts. The presented approach can already be used in operational forecasting systems for improving deterministic forecasts, or it can be developed further and integrated into a variational assimilation approach based on ensemble runs.

*Data availability.* TOPAZ4 ocean forcing data and Sea ice deformation data is publicly available at the Copernicus Marine Cervices portal:

  – https://resources.marine.copernicus.eu/product-detail/ARCTIC_ANALYSIS_FORECAST_PHYS_002_001_a/

  – https://resources.marine.copernicus.eu/product-detail/SEAICE_GLO_SEAICE_L4_NRT_OBSERVATIONS_011_006/

ECMWF atmosphere forcing data is available on the ECMWF website: https://www.ecmwf.int/en/forecasts/datasets.

neXtSIM code used in the present manuscript is not publicly available.

The forecasts are available per request.

*Author contributions.* This work is based on an original idea of PR. AK, PR and YY designed the methodology and AK carried out the simulation experiments and the analyses. EO, TW, PR and AK developed the neXtSIM model code. AK prepared the manuscript with contributions from all co-authors.

*Competing interests.* The authors declare that they have no conflict of interest.

*Acknowledgements.* This study was supported by the following projects: ImpSim funded by the Service hydrographique et océanographique
de la marine (SHOM), MOIRA funded by the European Space Agency (ref.no. 4000129593/19/I-DT), MUSIC funded by the Research Council of Norway (ref.no. 325292 - FORSKER21), and Nansen Legacy funded by the Research Council of Norway (grant no. 276730). The completion of this publication have been supported with institutional basic funding from the Nansen Center (RCN project # 218857). We thank the Copernicus Marine Services for providing the sea ice drift and deformation data product. We also thank Sylvain Bouillon for the numerous discussions that motivated this study.

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

## Appendix 1. Inverse observational operators

### Inverse observational operator for damage $H'_d$

The operator $H'_d$ (Eq. 20) is established from the following considerations. A true relationship between deformation and model variables is multivariate and involves nonlinear dependencies on the external forcing: for example, even fully-damaged ice will not deform without winds or currents. Satellite observations and previous studies with the neXtSIM model show that the values of ice deformation follow a log-normal distribution (Marsan et al., 2004; Rampal et al., 2019). Our simulations (Fig. 12, A) show that a linear relationship can be established between damage and total deformation in log-log space, i.e.,

$$\log_{10}(k_1 + 1 - d) = k_2 + k_3 \log_{10}(\varepsilon_{tot}) \tag{25}$$

where $k_2$ and $k_3$ are linear regression coefficients and $k_1$ is a small offset to prevent damage from getting too close to 1 (a critical value that damage should never reach in progressive damage models; Amitrano et al., 1999).

The coefficients $k_1$, $k_2$ and $k_3$ are found empirically following two steps. First, both the damage ($d$) and the simulated deformation ($\varepsilon_{tot}$) are taken from the truth run and the preliminary parameters are found by the minimisation in Eq. 19. The scatter-plot on Fig. 12, B compares the simulated damage (in $\log_{10}(1-d)$ space) with the damage reconstructed from the simulated $\varepsilon_{tot}$ using the inverse operator, showing reasonable agreement despite the aforementioned factors. The maps in Fig. 13 compare the simulated deformation, simulated damage and damage reconstructed from simulated deformation using $H'_d$, and show good agreement for large values of damage. In the range of low deformations, we note however that the agreement is not as good.

In the second step, the deformation is taken from satellite observations ($\varepsilon^o_{tot}$) and damage is derived by the inverse operator using the preliminary coefficients $d^o_1 = H'_d(\varepsilon^o_{tot})$. Comparison of the probability distribution functions (PDFs) of the simulated damage ($d$, blue area on Fig. 14) and the reconstructed damage ($d^o_1$, red line on Fig. 14) show deviations of PDFs due to the initial differences between the simulated and observed frequency distributions of deformation that result from varying

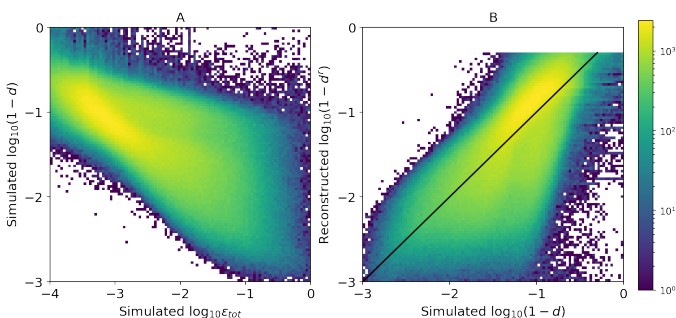

**Figure 12.** Comparison of simulated total deformation $\varepsilon_{tot}$, simulated damage $d$ and damage reconstructed from simulated deformation $d^r$ using Eq. 20. The black line on panel B shows 1-to-1 relation. Colours show density of points. Values of reconstructed damage below 0.5 (high log10 (1 - d′ )) are not shown on panel B.

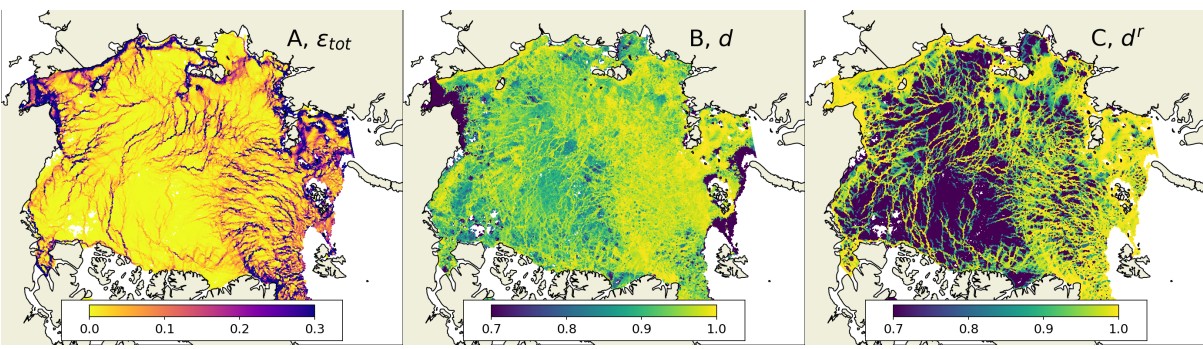

**Figure 13.** Comparison of maps of simulated total deformation $\varepsilon_{tot}$, simulated damage $d$ and damage reconstructed from simulated deformation $d^r$ for $5^{th}$ January 2020.

integration time of satellite observations, noise in observations, and uncertainties in simulated ice drift. The coefficients $k_1$, $k_2$ and $k_3$ are updated in a semi-automatic multivariate minimisation of the difference between the PDFs and $d_2^o$ is computed using the updated $H_d'$ (black line on Fig. 14). Values of the $H_d'$ parameters after the two steps are given in Table 2, which shows that the histogram fitting changes the values only marginally.

**Table 2.** Parameters of $H_d'$ operator after two steps of tuning.

|         | $k_1$ | $k_2$ | $k_3$ |
|---------|-------|-------|-------|
| $d_1^o$ | 0.05  | -2.7  | -0.9  |
| $d_2^o$ | 0.01  | -3    | -1.2  |

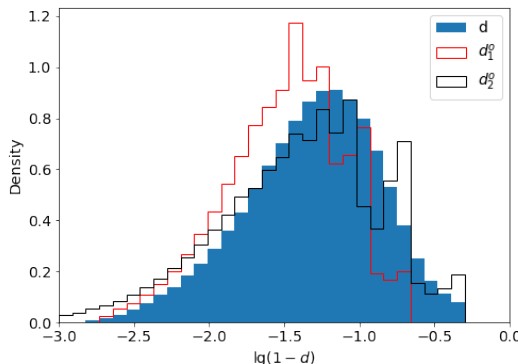

**Figure 14.** Comparison of probability density functions of simulated damage ($d$) and damage reconstructed from CMEMS observations of deformation using the first ($d_1^o$) and the second ($d_2^o$) sets of coefficient for $H_d'$.

**Inverse observational operator for concentration $H_A'$**

The simpler form of operator $H_A'$ (Eq. 21) can be justified by the fact that decrease in concentration purely due to divergence can be given by an integral of the divergence rate ($\varepsilon_{div}$) over time, therefore the coefficient $a_1$ relates to the integration time. However, in Eq. 21, the concentration of older sea ice is a function of $\varepsilon_{tot}$ assuming that ice breaks and becomes weaker due to both convergence, divergence and shear. Therefore, Eq. 21 is not a strict relation and the parameter $a_1$ is derived empirically in
the sensitivity experiments. An optimal value of $a_1$ is selected to keep both quality metrics $A_{MCC}$ and $KS$ as low as possible (see Sec. 4.4 and Fig. 8). Note, that unlike Eq. 19, the optimisation is performed here in the space of observed and simulated total deformation: $\max_{H'}(A_{MCC}(\varepsilon_{tot}, \varepsilon_{tot}^o))$.

The comparison of simulated deformation, simulated concentration and reconstructed concentration is provided on scatterplots (Fig. 15) and on maps (Fig. 16). The overall agreement between simulated and reconstructed concentrations is good, but
the simulated concentration is low only in areas where the divergence is high, whereas the reconstructed concentration is lower also in the areas with strong convergence or shear and represents weaker sea ice.

**Appendix 2. Using maximum cross-correlation for comparing deformation fields**

The satellite derived sea ice deformation field is a rasterized product with size $900 \times 900$ pixels and with spatial resolution of 10 km. The simulated sea ice deformation is computed on the model triangular mesh using contour integrals of the ice
drift velocity (for details of deformation computation see, e.g. Rampal et al., 2016). Then the deformation field is rasterized - resampled from the model mesh to the grid of the satellite observations using a nearest-neighbour method. Comparison between the two gridded deformation products (e.g., derived from satellite observation and obtained from a simulation) is performed by computing maximum cross correlation (Brunelli, 2009) as explained below.

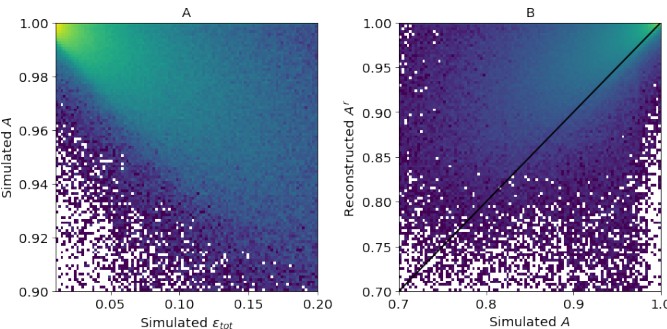

**Figure 15.** Comparison of simulated total deformation $\varepsilon_{tot}$, simulated concentration $A$ and concentration reconstructed from simulated deformation $A^r$ using Eq. 21. The black line on panel B shows 1-to-1 relation.

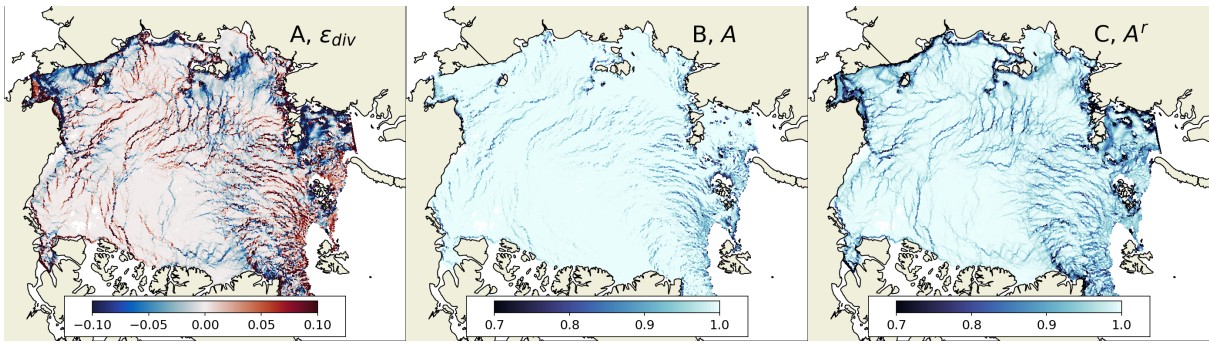

**Figure 16.** Maps of simulated divergence $\varepsilon_{div}$, concentration $A$ and concentration reconstructed from simulated total deformation $A^r$ for $5^{th}$ January 2020. The map of total deformation used for reconstruction of $A^r$ is shown on Fig. 13.A.

The grid of the tested product is split into smaller square matrices of size $N \times N$ pixels (called the template), the grid of the reference product is split into slightly larger matrices (with size $K \times K$ pixels, called the image) with the same geographic location of the centre of the corresponding matrices. The cross-correlation matrix (CCM) between the template and the image is computed as follows (see also scheme on Fig. 17):

$$R(x,y) = \frac{\sum_{x',y'}(T'(x',y') \cdot I'(x+x',y+y'))}{\sqrt{\sum_{x',y'}T'(x',y')^2 \cdot \sum_{x',y'}I'(x+x',y+y')^2}}$$

$$T'(x,y') = T(x',y') - 1/(w \cdot h)\sum_{x'',y''}T(x'',y'')$$

$$I'(x,y') = I(x',y') - 1/(w \cdot h)\sum_{x'',y''}I(x'',y'') \tag{26}$$

where $T$ is the template and $I$ is the quantized image, $x'$ and $y'$ are column/row coordinates of the centre of the image, $x$ and $y$ are column/row coordinates of the CCM.

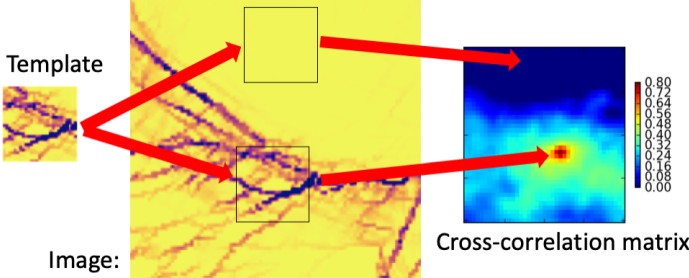

**Figure 17.** Scheme of computing the maximum cross correlation.

The maximum value from the CCM is used as a measure of similarity between the template and the image. The difference between the size of the template and the image ($(K - N)/2$) defines the tolerance of geographical misplacement of the tested and reference deformation fields. In our case we used template with size $K = 30$ pixels and image with size $N = 36$ pixels, meaning that a misplacement of 30 km was tolerated.