# Peer review of "Towards improving short-term sea ice predictability using deformation observations"

_The Cryosphere, 2022_

## Referee Comment (RC2)

Author: A. Korosov, P. Rampal, Y. Ying, E. Olason, T. Williams

Title: Towards improving short-term sea ice predictability using deformation observations

In this paper, damage and concentration – assumed to be a function of satellite-derived total sea ice deformation – are assimilated using a simple nudging method in the Brittle-Bingham-Maxwell (BBM) model – an off-spring of the EB/MEB model – for short-term sea ice forecast of Linear Kinematic Features. Results show that assimilation of sea ice concentration improves the skill of a short-term forecast for up to five days lead-time in comparison with persistence or a free simulation. The assimilation of damage on the other hand does not improve the skill of the forecast. It is argued that that the reason for the lack of sensitivity to damage assimilation is because the healing of ice (that reduces the damage) has a timescale that is too fast (~24 hrs) compared with the desired lead time of the forecast (O(days)).

The paper is well written. The organisation and depth of the paper however is lacking. The text refers to a non-existent appendix on several occasions; the sensitivity to assimilation parameters consists of four 1 or 2-line paragraphs; there are multiple unsubstantiated statements; the results section presents inconsistent results (sea ice concentration larger than one) and focuses on small scale features in order to justify broad conclusions. The authors state that neXtSIM is not ready to assimilate deformation yet and that the paper is a proof-of-concept. There is no problem with this approach, but the authors have pushed this paper out for review much too early. I recommend that the paper be rejected for the moment and that authors be encouraged to resubmit later, a substantially different version of the paper that addresses (not rebut) the comments below. I am recommending a "reject with encouragement to resubmit" simply to give the author time to properly address the comments.

**Overstatements**:

1- Line 44: "In a recent model intercomparison paper (Bouchat et al., 2021), only one model, neXtSIM (neXt Generation Sea Ice Model, Bouillon and Rampal, 2015a; Rampal 45 et al., 2016), proved to be capable when run at the same spatial resolution as the available observations (i.e. ~ 10km) to simulate the observed probability distribution, spatial distribution and fractal properties of sea ice deformation." This is incorrect. One 10-km McGill run was able to reproduce the PDF of divergence and multiple 10-km models were able to reproduce temporal scaling and spatio-temporal coupling. The only place where neXtSIM did perform better compared to other 10-km models is the spatial scaling.

2- Line 304: "The viscous-plastic (VP) rheology used in MITgcm is known to have a less realistic slower time evolution of LKFs (Hutter et al., 2018) than the BBM rheology in neXtSIM (Olason et al., 2022). As a result, the sea ice simulated by the BBM rheology has more rapid error growth (loses skill faster) due to the correctly resolved intermittent ice motion and localised ice deformation." If the intermittent motion is correctly resolved, why is the error growing faster? Intermittency does not come from the brittle parameterization in the EB/MEB/BBM. All models participating in SIREX (VP, EVP,

EB) show intermittency (Bouchat et al., 2021). The source of intermittency in observed deformation is still an unresolved issue. The authors must clarify what they mean by "intermittency. If the temporal scaling exponent is used to discuss the LKFs intermittency, this is incorrect. If instead the authors are referring to LKF growth rate and lifetime, this is also incorrect. In SIREX2, it is shown that no apparent link is present between the LKF growth rates, lifetimes and the temporal scaling/multi-fractal parameters. The intermittency is revealed by the quadratic nature of the structure function.

3- Line 209: "Thus, due to its rheology, neXtSIM is able to extrapolate and create realistic connections between the observed and assimilated pieces of LKFs." This is not demonstrated in the manuscript. I believe any rheology that assimilates sea ice contration (A) will show LKFs in line with observations. See Major Points below.

Major Points:

1- On two occasions, the authors are referring to an Appendix that is not included in the paper. The appendix must be included.

2- Equation 7-10: Sea ice concentration (A) increases in convergence (until A=1) and decreases in divergence; sea ice concentration can also increase or decrease in shear. A single dependency of A on eps_tot is therefore missing events where convergence is present along LKFs. The damage (d) dependency on eps_tot is more realistic, but the authors argue that assimilation of A is useful to increase predictive skill, whereas d is not. See below for more on this topic. The single dependence of A on (eps_tot) must be justified.

3- Line 145: The value of a1 in F_A(eps_tot) is found through sensitivity experiments using the same BBM model. This is a circular argument. This functional dependence must be derived from sea ice concentration and total deformation derived from passive microwave and Sentinel. I believe the author will find that the functional dependency is not a simple linear relationship. The author must at least show this relationship from observations and acknowledge the simplicity/caveat of the approach.

4- Fig 5a: The skill of the model is assessed using the fraction of points where the correlation between observed and simulated deformation is significant. Statistically speaking, there will always be some points that will remain significantly correlated. The statistical significance of the signal must be shown in the figure. I also see no spatial structure in the regions of high correlations which suggest that the high-correlation points are just random events.

5- Fig 5b: I would have expected that the root mean square difference of the forecast run would asymptote to the free run. The fact that it does not is suspicious. This must be explained.

6- The discussion of the sensitivity of the forecast on epsilon_min and the weighting factor w_d appears in two one-line paragraph. I suggest removing them, or a more in-depth discussion should be provided.

7- Figure 6. The constant a1 is negative. This means that the sea ice concentration is larger than 1 (see Equ 10 and since eps_tot > 0). This is not physical. The results in the figure cannot be correct.

8- I am assuming that the sign of a1 is incorrect. If so, A = F_A(eps_tot) = 1 – a1 * eps_tot ~=0.76 for a1 ~= -1.2 (Line 232) and eps_tot ~= 0.2 (Fig 3). For A=0.76, P* is scaled down two orders of magnitude and the ice has no strength. Any rheological model where the ice strength is set to nearly zero along a line (an observed LKF in this case) will deform along that line – this can be tested simply. The author instead argue that the brittle rheology is key to the correct simulated location of the LKFs. This is another unsubstantiated statement.

9- Results: The simulated concentration and thickness fields after assimilation should be presented. Reading from the deformation fields and the a1 constant derived from sensitivity experiments, we should see concentration of ~0.8 along LKFs, something that is not accord with RGPS observations at 10km scale resolution. I suspect assimilating damage would help producing more realistic fields. The authors give reasons for why damage assimilation is not successful, but those are not convincing. See below for more details on this topic.

10- It is argued that the damage does not increase predictive skill of LKFs because ice heals too rapidly (~24 hours). In the real world, ice heals through thermodynamic processes on much longer time scale. This choice of short healing time scale must be justified. Perhaps this is the cause of the lack of predictive associated with the assimilation of the damage.

11- The error is shown as ¼ sigma. This is highly unusual. Typically, one would show an error envelope equal to one sigma (four times larger than what is shown in Figure 5).

Minor Points:

1- Line 23: "Under external forcing the ice deforms primarily as an elastic material." Most deformations in the pack ice are plastic and occurs along LKFs. This sentence also contradicts the next sentence.

2- Line 26: "…start deforming along multiple narrow and elongated cracks formed and does so until these later refreeze". Or when the load (winds) on the ice changes. This should be added.

3- Line 48: :"… the exact timing and position of strong deformation zones, or LKFs, is not yet predicted precisely". The exact position of LKFs will never be located precisely because it depends on unresolved weaknesses within the ice pack. What we can hope to reproduce is the timing, the orientation of the LKFs with respect to the large-scale forcing and their statistical distribution (width, length, density, angle of fracture, etc.). This should be corrected. This is another sentence that suggests incorrectly that BBM could eventually simulate LKFs position correctly.

4- Line 93: "The observed variables vo (damage and concentration) is computed…". "Damage" is not observed. The word "is" should read "are".

5- Line 94: Sometimes, the Greek lowercase epsilon symbol is used and sometimes the lunate epsilon symbols is used. The author needs to choose only one form for consistency. See Line 94 and Equations 6-7 for examples. This needs to be corrected everywhere.

6- Line 102: It is said that CNEMS has a temporal resolution of 12 hours on Line 102; and it is said in the next sentence that it is "observed nearly every day". This sounds contradictory. This should be clarified. See Line 115 as well where 24 hours is specified.

7- Line 105: "The model is forced with the European Center for Medium-Range Weather Forecasts (ECMWF)". The version of ERA should be specified: e.g. ERA5, ERA-interim, etc.
8- Line 117: "total" should read "total deformation".
9- Line 203: The units of eps_tot must be given the first time it is introduced (Table 1, Line 197). At present it is only given on Line 203.
10- Line 235: eps_tot and divergence is used interchangeably; yet they are very different. One includes shear the other not.

Bruno Tremblay
McGill University

---

## Author Comment (AC1)

**Replies to Reviewer 1**

This manuscript describes how including sea ice deformation derived from a satellite data product of sea ice drift may improve short sea ice predictive skills of an Arctic sea ice model. The main "trick" is to connect derived deformation to scalar model variables with some "memory", i.e ice concentration and damage, as assimilating the deformation or drift directly is known to be problematic as this information contains little memory and is usually lost within time steps of the model. The data assimilation (DA) itself is a simple re-initialisation scheme for ice concentration and damage. The authors speculate how their findings can be used in more sophisticated DA systems.

The topic is interesting and important as sea ice forecasts are thought to become more and more relevant for Arctic shipping and exploration. There are some issues with the manuscript that require careful rewriting and even repeating the experiments, so that I cannot recommend publication in the present form.

We appreciate the reviewer's constructive comments. As requested by the reviewer, more experiments for evaluating predictability of sea ice deformation were run. We have also improved the clarity of the mathematical formulation of the problem. The text below, which describes these runs and defines the terminology, is added to the Methodology section. See also Fig. 1 for further explanation.

Let **x** denote the model state variables (e.g., concentration, damage, etc.). Experiments start from 1 December 2020  $(t_0)$  and last for two months.  $\mathbf{x}_{t_0}$  is the initial condition.  $M_{t_n \to t_{n+1}}$  is the non-linear model (neXtSIM) to propagate state from time  $t_n$  to  $t_{n+1}$ .

Let  $\mathbf{y}$  denote the observations (total deformation rate), which is related to the model state variables through  $\mathbf{y}_t = H(\mathbf{x}_t)$ , where H is the observation operator. Real satellite observations  $\mathbf{y}_t^o$  are available throughout the test period. Although the deformation rate is derived from sea ice drift, derived from Radarsat-2 SAR images, we call them "observations of deformation" as opposed to "simulation of deformation" by neXtSIM.

In the first experiment a verifying "truth run" is generated:

$$\mathbf{x}_t^{\mathrm{tr}} = M_{t_0 \to t}(\mathbf{x}_{t_0})$$

The period before 1 January 2021 is used as a spin-up time and the data from  $\mathbf{x}_t^{\text{tr}}$  is not used.

Then four sets of 10-days forecasts are ran every day:

**1. Forecasts initiated from truth:**

$$\mathbf{x}_{t_1 \to t}^{\mathrm{tr}} = M_{t_1 \to t}(\mathbf{x}_{t_1}^{\mathrm{tr}}) + \boldsymbol{\psi}_t,$$

where  $\psi_t$  is a random noise due to uncertainties in model numerics that cause the forecasts run to differ from the truth run. These forecasts are evaluated by computing the error in observation space:

$$\varepsilon_T = \left\langle H(\mathbf{x}_{t \to t+\delta t}^{\mathrm{tr}}) - \mathbf{y}_{t+\delta t}^{tr} \right\rangle,$$

where  $\langle \cdot \rangle$  denotes averaging over the different forecasts starting from  $t_1, t_2, \ldots, t_n$ , (i.e.  $\langle e_{t \to t+\delta t} \rangle = \sum_{t=t_1}^{t_n} e_{t \to t+\delta t}$ , then plotted w.r.t. lead time  $\delta t$ ).

**2. Forecasts without data assimilation:**

$$\mathbf{x}_{t_1 \to t} = M_{t_1 \to t}(\mathbf{x}_{t_1}) + \boldsymbol{\psi}_t.$$

The first forecast is initiated from  $t_0$ , subsequent forecasts are initiated from the outputs of the previous forecasts. The forecasts initiated during the spin-up period are not used. The forecasts after 1 January are evaluated against truth:

$$\varepsilon_t^B = H(\mathbf{x}_t) - \mathbf{y}_t^{tr} ,$$

and against real observations:

$$\varepsilon_t^O = H(\mathbf{x}_t) - \mathbf{y}_t^o.$$

During the spin-up period  $\varepsilon_t^B$  grows and reaches its saturation level  $\varepsilon_B$ , which we consider to be the climatological level for this error. Since the forecasts without data assimilation don't see real data, the error  $\varepsilon_t^O$  averaged over one month ( $\varepsilon_O$ ) can also be considered as the climatological level.

**3. Forecasts with assimilation of synthetic data:**

$$\mathbf{x}_{t_1 \to t}^{as} = M_{t_1 \to t}(\mathbf{x}_{t_1}^{as}) + \boldsymbol{\psi}_t,$$

where  $\mathbf{x}_{t_1}^{as}$  is the analysis of synthetic observations from the truth run and the forecasts without assimilation performed at  $t_1$ :  $\mathbf{x}_{t_1}^{as} = A(\mathbf{x}_{t_1}, \mathbf{y}_{t_1}^{tr}; H', \mathbf{w})$ . In the assimilation scheme in this paper, we use the inverse operator H' to compute model state (concentration and damage) from the observed deformation,  $\mathbf{x}_t = H'(\mathbf{y}_t)$  (see Eqs. 9 and 10 for how H'is constructed) and  $\mathbf{w}$  are the tuning parameters (see Eqs. 11 - 13 for how A is constructed). These forecasts are evaluated with:

$$\varepsilon_S = \left\langle H(\mathbf{x}_{t \to t+\delta t}^{as}) - \mathbf{y}_{t+\delta t}^{tr} \right\rangle$$

4. Forecasts with assimilation of real satellite data:

$$\mathbf{x}_{t_1 \to t}^{ar} = A(\mathbf{x}_{t_1}, \mathbf{y}_{t_1}^o; H', \mathbf{w}),$$

evaluated with:

$$\varepsilon_A = \left\langle H(\mathbf{x}_{t \to t+\delta t}^{ar}) - \mathbf{y}_{t+\delta t}^{o} \right\rangle.$$

Predictability is defined as the time at which a forecast error reaches a background level [Zhang et al., 2019]. Since the errors of the forecasts without assimilation  $\varepsilon_B$  and  $\varepsilon_O$  are at their respective saturation levels, we assume them to be the background levels for the forecasts with assimilation. Therefore, in a perfect-model scenario (forecast with initialisation from truth) the intrinsic predictability is the time  $\delta t$  when  $\varepsilon_T \approx \varepsilon_B$ . The practical predictability is the time  $\delta t$  when  $\varepsilon_S \approx \varepsilon_B$ . Similarly, in the case of assimilation of real observations the practical predictability is time  $\delta t$ when  $\varepsilon_A \approx \varepsilon_O$ .

Figure 1: Scheme of experiments (left) and scheme of errors (right). The truth run is shown by solid blue line, observations - by solid red line. The forecasts initiated from the truth - dashed blue line; without assimilation - solid green line; with assimilation of synthetic observations - dashed green line; with assimilation of real satellite observations - dashed red line. Grey lines show spin-up period for the truth and the no-DA forecasts. On the scheme of errors the lines are coloured as follows: red - evaluation against satellite observations, green - evaluation against synthetic observations, blue - evaluation against the "truth run". Solid lines show climatological level, dashed - average over forecasts. Vertical lines  $P_T$ ,  $P_S$  and  $P_A$  indicate potential predictability, practical predictability for synthetic DA and practical predictability for real DA, correspondingly.

**Main points**

1. According to Fig2 and the text, the data assimilation scheme uses observational data from the period between t0 and t1 to initialise the model at t0; more generally the model is initialised at t(n) with data from between t(n) and t(n+1) when in a realistic system, the data is not yet available. The difference between t(n) and t(n+1) is 24h. Then the same data set is used to evaluate the result of the assimilated model, i.e. the evaluation of the "forecast" on the first day is with the same data as the data set that is used to initialise the forecast. Since in the simple scheme, the initialisation is done neglecting the corresponding models value entirely (weight 1 in Section 3.4, 4.1), this comparison only

shows how well the model persists the initial conditions. Not surprisingly, the model/data "agreement" is quite good on the first day and quickly deteriorates on day 2-5. A proper scheme would use data from t(n-1) to t(n) to initialise at t(n) and compare to data at t(n), t(n+1), etc. As long as this is not change, the first day of the "forecast" cannot be used for any analysis and shouldn't even be called a forecast. The model runs between t(n) and t(n+1).

All the previous and many additional experiments were re-run with assimilation of historical data only. The deformation is now computed from observations of drift at t(n-1) and t(n), then the analysis is computed and the model is initialised at time t(n), then the forecast is compared with deformation computed from t(n) to t(n+1), from t(n+1) to t(n+2), etc. The new experiments confirm that assimilation of data from the day before improves the accuracy of the deformation forecast both in case of synthetic and real data. New results are shown on Fig. 2 and are added to the Results section.

Figure 2: Errors of forecasts. Lines are coloured according to the same scheme as on Fig. 1.

2. Terminology and language. The use of established terminology is rather liberal in the manuscript. As far as I know (but I may be wrong), the terms prediction, predictive skill, predictability, potential predictability have well defined meanings (I haven't heard of "prediction skill"), and the terms should be used when describing the experiments, otherwise it is hard to relate the work to other DA publications. Similarly, the DA scheme is described as "nudging", whereas it is a weighted re-initialisation scheme (according to eq11), but then the weights are always 1 or 0, so there is no weighting in this manuscript. "Nudging" implies a term in an evolution equation d v /dt = some terms + nudgingParamter \* ( $v_o - v$ ), where  $v_o$  is the observation. Many smaller problems of similar nature can be found in the text. I marked some of them in the list below.

The terminology has been changed according to the reviewer comments. The term "nudging" is replaced with "direct insertion" [Stanev and Schulz-Stellenfleth, 2014]. In the new experiments we tested different values of the weights [0, 0.5, 1].

3. In data assimilation, one can expect that including additional information will improve the result. Therefore, comparing an assimilated model to a free run makes little sense. Essentially the free run in Fig3 is a 22-day forecast that hasn't seen any new data in 22 days. As noted before, comparing the model to observations that have been used in the assimilation cannot say much about the "success" of assimilation. Anything but any small improvement would just be a failure. Similar "mistakes" have been made before, e.g. doi:10.3189/2015AoG69A740

In general, one would have a ctrl-simulation with an established DA scheme and then add new data or new methods and compare the improvements over the ctrl-simulation. This is done in section 4.2,

but it would be more interesting to see, if the addition of deformation data to an existing DA system (which may already assimilate ice concentration or even thickness) would improve the predictive skill. The authors present their work as a "proof of concept", but the evidence they provide does not help in evaluating, if these additional data help in a realistic system, because the framework is so different.

In the new experiments we evaluate the forecasts on independent data which was not used for assimilation. As explained above, the difference between the forecasts without assimilation and the observations provides the background error. The error of the forecast with assimilation is compared to the background error for evaluating practical predictability. Our previous experiments with assimilation of sea ice concentration [Williams et al., 2021] showed that ice drift is not significantly affected by assimilation. In that sense the ctrl-simulation with assimilation of, e.g., ice concentration is almost equal to the forecasts without assimilation.

4. When introducing new data and constraining new model variables it is good practice (also in sea ice data assimilation) to test schemes and types of data in twin experiments, where a free run produces "observations", a subset of which is used for assimilation leaving the remaining data for validation. This has been done a lot especially in anticipation of new data (e.g. doi:10.1029/2006JC003786). Here, one could have at least held back some of the observations to be used for model validation. This makes it impossible to check if the DA actually improves the state also away from the observations. Instead, one can only make statements about the plausibility of the solutions outside the areas covered by observations (by no mean can the authors claim, that the LKFs are "corrected" outside of the data coverage).

The following twin experiments were run as explained above:

- Truth run,
- Forecast initialised from truth,
- Forecast without assimilation,
- Forecast with assimilation of synthetic data,
- Forecast with assimilation of real data.

As explained on Fig. 1 and 2 these experiments allowed to evaluate the practical predictability on synthetic observations and real observations.

Satellite observations already have large gaps and artificially reducing the coverage even further will decrease the ability of neXtSIM to connect the pieces of LKFs. Luckily we have a few cases when the coverage of test data exceeds the coverage of assimilated data, and we can test our hypothesis that LKFs are improved also outside the area of assimilation as shown on Fig. 4 in the manuscript. In the new set of experiments we use independent data for testing and can confirm that MCC increases also outside the area of assimilation. Nevertheless, due to lack of sufficient cases of confirmed good extrapolation, the statement on correction of LKFs in the entire basin is rephrased as follows:

"... it illustrates that even if data insertion is spatially limited by satellite observations (or even very localized in high deformation zones) it can realistically extrapolate the deformation pattern by connecting the elements of linear kinematic features."

Experiments with spatially limited assimilation of synthetic data also confirm that accuracy of LKFs is increased not only in the area of assimilation, as shown on Fig. 3.

5. A key point of the procedure is how the deformation derived from satellite ice drift data is connected to the model variables concentration and damage. The derivation of this empirical connection is moved to the "supplemental material/Appendix I", which is not part of the manuscript, nor can it be found online.

Appendix I was not included in the first submission due to a technical problem. It will be included in the manuscript.

**Minor issues, typos, suggestions, some related to the above points.**

page 1

12: due to the lack - > in the next sentence there are observations to be assimilated. Please rewrite.

---

## Author Response (AR1)

**Replies to Reviewer 1**

This manuscript describes how including sea ice deformation derived from a satellite data product of sea ice drift may improve short sea ice predictive skills of an Arctic sea ice model. The main "trick" is to connect derived deformation to scalar model variables with some "memory", i.e ice concentration and damage, as assimilating the deformation or drift directly is known to be problematic as this information contains little memory and is usually lost within time steps of the model. The data assimilation (DA) itself is a simple re-initialisation scheme for ice concentration and damage. The authors speculate how their findings can be used in more sophisticated DA systems.

The topic is interesting and important as sea ice forecasts are thought to become more and more relevant for Arctic shipping and exploration. There are some issues with the manuscript that require careful rewriting and even repeating the experiments, so that I cannot recommend publication in the present form.

We appreciate the reviewer's constructive comments. As requested by the reviewer, more experiments for evaluating predictability of sea ice deformation were run. We have also improved the clarity of the mathematical formulation of the problem – the methodology section has been completely re-written.

**Main points**

**1. According to Fig2 and the text, the data assimilation scheme uses observational data from the period between t0 and t1 to initialise the model at t0; more generally the model is initialised at t(n) with data from between t(n) and t(n+1) when in a realistic system, the data is not yet available. The difference between t(n) and t(n+1) is 24h. Then the same data set is used to evaluate the result of the assimilated model, i.e. the evaluation of the "forecast" on the first day is with the same data as the data set that is used to initialise the forecast. Since in the simple scheme, the initialisation is done neglecting the corresponding models value entirely (weight 1 in Section 3.4, 4.1), this comparison only shows how well the model persists the initial conditions. Not surprisingly, the model/data "agreement" is quite good on the first day and quickly deteriorates on day 2-5. A proper scheme would use data from t(n-1) to t(n) to initialise at t(n) and compare to data at t(n), t(n+1), etc. As long as this is not change, the first day of the "forecast" cannot be used for any analysis and shouldn't even be called a forecast. The model runs between t(n) and t(n+1).**

All the previous and many additional experiments were re-run with assimilation of historical data only. The deformation is now computed from observations of drift at t(n-1) and t(n), then the analysis is computed and the model is initialised at time t(n), then the forecast is compared with deformation computed from t(n) to t(n+1), from t(n+1) to t(n+2), etc. The new experiments confirm that assimilation of data from the day before improves the accuracy of the deformation forecast both in case of synthetic and real data. New results are shown on Fig.6 in the revised manuscript in the Results section.

**2. Terminology and language. The use of established terminology is rather liberal in the manuscript. As far as I know (but I may be wrong), the terms prediction, predictive skill, predictability, potential predictability have well defined meanings (I haven't heard of "prediction skill"), and the terms should be used when describing the experiments, otherwise it is hard to relate the work to other DA publications. Similarly, the DA scheme is described as "nudging", whereas it is a weighted re-initialisation scheme (according to eq11), but then the weights are always 1 or 0, so there is no weighting in this manuscript. "Nudging" implies a term in an evolution equation d v /dt = some terms + nudgingParamter * ( $v_o$ - v ), where $v_o$ is the observation. Many smaller problems of similar nature can be found in the text. I marked some of them in the list below.**

The terminology has been changed according to the reviewer comments. The term "nudging" is replaced with "direct insertion" [Stanev and Schulz-Stellenfleth, 2014]. In the new experiments we tested different values of the weights [0, 0.5, 1].

**3. In data assimilation, one can expect that including additional information will improve the result. Therefore, comparing an assimilated model to a free run makes little sense. Essentially the free run in Fig3 is a 22-day forecast that hasn't seen any new data in 22 days. As noted before, comparing the model to observations that have been used in the assimilation cannot say much about the "success" of assimilation. Anything but any small improvement would just be a failure. Similar "mistakes" have been made before, e.g. doi:10.3189/2015AoG69A740**

**In general, one would have a ctrl-simulation with an established DA scheme and then add new data or new methods and compare the improvements over the ctrl-simulation. This is done in section 4.2, but it would be more interesting to see, if the addition of deformation data to an existing DA system (which may already assimilate ice concentration or even thickness) would improve the predictive skill. The authors present their work as a "proof of concept", but the evidence they provide does not help in evaluating, if these additional data help in a realistic system, because the framework is so different.**

In the new experiments we evaluate the forecasts on independent data which was not used for assimilation. As explained above, the difference between the forecasts without assimilation and the observations provides the background error. The error of the forecast with assimilation is compared to the background error for evaluating practical predictability. Our previous experiments with assimilation of sea ice concentration [Williams et al., 2021] showed that ice drift is not significantly affected by assimilation. In that sense the ctrl-simulation with assimilation of, e.g., ice concentration is almost equal to the forecasts without assimilation.

**4. When introducing new data and constraining new model variables it is good practice (also in sea ice data assimilation) to test schemes and types of data in twin experiments, where a free run produces "observations", a subset of which is used for assimilation leaving the remaining data for validation. This has been done a lot especially in anticipation of new data (e.g. doi:10.1029/2006JC003786). Here, one could have at least held back some of the observations to be used for model validation. This makes it impossible to check if the DA actually improves the state also away from the observations. Instead, one can only make statements about the plausibility of the solutions outside the areas covered by observations (by no mean can the authors claim, that the LKFs are "corrected" outside of the data coverage).**

The following twin experiments were run as explained above:

- Truth run,
- Forecast initialised from truth,
- Forecast without assimilation,
- Forecast with assimilation of synthetic data,
- Forecast with assimilation of real data.

As explained on Fig. 1 in the manuscript these experiments allowed to evaluate the practical predictability on synthetic observations and real observations.

Satellite observations already have large gaps and artificially reducing the coverage even further will decrease the ability of neXtSIM to connect the pieces of LKFs. Luckily we have a few cases when the coverage of test data exceeds the coverage of assimilated data, and we can test our hypothesis that LKFs are improved also outside the area of assimilation as shown on Fig. 4 in the manuscript. In the new set of experiments we use independent data for testing and can confirm that MCC increases also outside the area of assimilation. Nevertheless, due to lack of sufficient cases of confirmed good extrapolation, the statement on correction of LKFs in the entire basin is rephrased as follows:

"Our experiments illustrate that even if data insertion is spatially limited by satellite observations (or even very localized in high deformation zones) it can realistically extrapolate the deformation pattern by connecting the elements of linear kinematic features."

Experiments with spatially limited assimilation of synthetic data also confirm that accuracy of LKFs is increased not only in the area of assimilation, as shown on Fig. 5 in the manuscript.

**5. A key point of the procedure is how the deformation derived from satellite ice drift data is connected to the model variables concentration and damage. The derivation of this empirical connection is moved to the "supplemental material/Appendix I", which is not part of the manuscript, nor can it be found online.**

Appendix I was not included in the first submission due to a technical problem. It is included in the manuscript.

**Minor issues, typos, suggestions, some related to the above points.**

**page 1**
**l2: due to the lack $->$ in the next sentence there are observations to be assimilated. Please**

**rewrite.**

Rewritten as follows: "Short-term sea ice predictability is challenging despite recent advancements in sea ice modelling and new observations of sea ice deformation, that capture small-scale features (open leads and ridges) at kilometre scale."

**l8: deterministic forecasting with one member − > isn't that a homoioteleuton? A one-member system is always deterministic, or an ensemble with just one member, is not really an ensemble. Remove "with one member", or replace "with a single simulation"**

The sentence is rewritten as follows: "This proof-of-concept assimilation scheme uses a data insertion approach and forecasting with one member. We obtain statistics of assimilation impact over a long test period with many realisations starting from different initial times."

**l10: in 3–5 days horizon − > grammar?**

The sentence is rewritten as follows: Assimilation and forecasting experiments are run on synthetic and real observations in January 2021 and show increased accuracy of deformation prediction for the first 2 - 3 days.

**l13: reduction in: article missing, or replace by "reducing the", although this sentence is not very clear in general and could be improved**

The sentence was removed.

**l13: bigger role − > than what?**

The sentence was removed.

**l20: only − > mainly (sea surface tilt, momentum advection, and you cannot exclude small effects of floe-floe interaction) or "dominated"**

The sentence is rewritten as follows: "...the speed and direction of the drift are dominated by the atmospheric and ocean drag forces and by the Coriolis force."

**l23: brittle − > I don't think that you can say that. It's driven by complex non-Newtonian mechanics/dynamics, but "brittle" is just one aspect of it, and frankly, only a model for the behavior. Other models of sea ice motion exist (I don't mean numerical models). Please rewrite.**

Relation to brittle rheology is removed.

**page 2, l25: deforming − > just to illustrate my previous point, this deformation is NOT brittle, but plastic (no restoring force pushes the ice back into the initial state as for elastic behaviour). The brittle part is just the way failure is parameterised in nextsim (which I believe is a good model for this). I think that this general description of sea ice mechanics/dynamics needs to be "decoupled" from the specific model nextsim, that is being used in this manuscript.**

The simplified explanation of brittle rheology is rewritten as follows:

"First, under increasing external forcing the undamaged ice deforms primarily as an elastic material. Internal stresses gradually accumulate in the material until a failure criterion is reached, which corresponds to a limit when sea ice fractures, and then the ice starts deforming along the multiple narrow and elongated cracks, and does so until these later refreeze or when the load (winds and currents) on the ice changes."

We agree that brittle sea ice deformation is more general than a specific BBM rheology or a specific implementation in neXtSIM. In the paragraph near line 25 neXtSIM is not mentioned.

**l28: "Under divergent ice motion these cracks become open leads, significantly increasing ocean-air heat and mass exchange and modifying local atmospheric boundary layer and ocean mixed layer. Open leads are also key both for marine fauna survival," − > I agree with this, certainly on the local scale of the leads, but this is just a plausibility argument. I have not yet seen that this has been confirmed on large scale heat and mass exchange and budgets. Please give references, if you have them, otherwise marks this as a "plausible assumption".**

A relevant reference (Olason et al., 2021) is added.

**l37: observe − > isn't RGPS a data product derived from Radarsat on a 12.5km grid? I find "observe" in this context inappropriate. Please rewrite.**

There are several RGPS products: a gridded deformation product on 12.5 km resolution and a Lagrangian product with a variable mesh size with a nominal resolution of approx. 10 km. The sentence is rewritten as follows:

"The Radarsat Geophysical Processor System (RGPS) dataset was the first attempt to systematically observe sea ice drift and derive sea ice deformation on high spatial resolution (10 km) and with high frequency (3 days) over a long period of time (winters 1996–2016) (Kwok, 1998)."

**l41: (10 - 30 km) − > can be only a result of the "coarse" resolution, or do we really have "cracks" in the Arctic that are 10-30km wide. Those would be large stretches of either open water or vigorous deformation. I assume that the interpretation is important for the DA.**

Rewritten as follows:

The cracks appear on satellite-derived ice deformation products as narrow (10 - 30 km, depending on resolution of satellite data) and long (up to 1000 km) lineaments and are also called linear kinematic features (LKFs) (Kwok, 2001).

**l44: "only one model, neXtSIM" − > Supposedly this is put here to justify the decision to use neXtSIM. The statement is not incorrect, but it is not clear to me, what the authors would like to achieve with this statement. It does not help this paper in any way, because it hides the results of the Bouchat's paper, that other models have similar properties (at finer grid spacing and higher computational cost). Also doesn't the nextsim in Bouchat's paper use the MEB rheology instead of the BBM-rheology? I would rewrite as something like this (I tried to emphasise that this is a useful model for this study, i.e. does the job very well and is comparatively cheap):**

**In a recent model intercomparison paper (Bouchat et al., 2021), neXtSIM simulations (neXt Generation Sea Ice Model, Bouillon and Rampal, 2015a; Rampal 45 et al., 2016), ranked among the best for simulating the observed probability distribution, spatial distribution and fractal properties of sea ice deformation, even though it operates on a low resolution grid of 10km. All other comparable simulations used higher resolution and were hence more expensive.**

As pointed out by the second reviewer, some of the models were running at a comparable resolution with quite good results. Therefore, the text is rewritten as follows:

"In a recent model intercomparison paper (Bouchat et al., 2021) neXtSIM results ranked among the best for simulating the observed probability distribution, spatial distribution and fractal properties of sea ice deformation, even though it operates on a low resolution grid of 10 km. Analysis of spatial and temporal scaling (Fig. 13 in (Bouchat et al., 2021)) shows that the spatial structure function of neXtSIM matches the RGPS observations very well, whereas the temporal one is overestimated by $3 - 5$ %, probably indicating some overestimation of the intermittency by neXtSIM."

**l49: skill − > the skill?**

Corrected

**l51: observations − > the technical term for this is "potential predictability", which always excludes observations. Why not use that?**

Rewritten as follows:

"Mohammadi-Aragh et al. (2018) evaluated the potential predictability of LKFs using an ensemble of sea ice models all using a viscous-plastic rheology, but the practical predictability remains unknown."

**l55: "so the assimilation scheme needs to perform a cross-variable update from deformation to sea ice model variables." This is a common "problem" in DA and one would use a proper "observation operator" that maps the model variables to the observations. The dual operation then maps the model-data misfit back to increments of model variables. If you want to talk about "data assimilation", I suggest to use the proper language/terminology. Here you will (according to the abstract) do a nudging experiment (but it turns out to be re-initialisation in reality), which is strictly speaking not really data assimilation (although total valid as a method).**

The term "nudging" is replaced with "direct insertion", which, according to e.g. [Stanev and Schulz-Stellenfleth, 2014], is one of the data assimilation methods. The sentence is rewritten as follows:

"First, the "direct insertion" method operates in the model state space. However, the observed deformation is not a model prognostic variable, so an operator is used to convert deformation to the model variables. This operator is an

inverse of the observation operators used in data assimilation, since it maps from the observation space back to the state space."

**page 3: l79: "d is the ice damage." maybe it makes sense to clearly state the mean of "d", d=0 is entirely intact and 1 is entirely damaged (which I assume here from the equation) or vice versa. Previous publications use contradicting definitions.**

Explanation is added: "(with $d = 0$ being completely undamaged ice)."

**page 4: eq5, what is "P"?**

Added after Eq 5.:

"where $P$ is a constant scaling parameter for the ridging threshold to parameterise $P_{max}$, following the results of Hopkins (1998), and $h$ is thickness."

**l101: "12 hours frequency" $->$ 12h is not a frequency, but a period. The frequency is: one record in 12h.**

Rewritten as follows:

"Ice drift is computed from pairs of images separated by approximately 24 hours and the product is delivered every 12 hours."

**page 5, l110: "reach an equilibrium" in 30 days is hard to believe, usually one would expect a seasonal cycle at least, unless the sea ice models of TOPAZ4 and nextsim are identical (which they are not, I assume). But does the equilibrium matter?**

The entire description of experiments is rewritten.

**Figure 1 is not really necessary.**

The entire description of experiments is rewritten and a scheme with explanations of several experiments becomes necessary.

**l113: this looks like the scheme uses data from the future to correct the model? Does that make any sense? I would expect to update the model variable at t(n) with data collected over t(n-1) (or earlier) to t(n). See also main points above.**

The experiments were re-run to validate the results on independent data.

**l119: in the previous work of (e.g. (Bouillon and Rampal, 2015b)). $->$ fix parentheses**

Corrected.

**Fig2, caption: Eps,d and A $->$ use proper symbols as in figure.**

Figure 2 is removed.

**page 6: l125 The "Appendix" should be in the same file as this text, right? Supplementary material is separate. What do we have here? On the TC-web page I cannot find any supplementary material, so that "Appendix I" is missing for now.**

Unfortunately, the Appendix was not added due to technical reasons. It is added in the revised manuscript.

**eq9: that would be $1 - 10^{k_2} * \varepsilon_{tot}^{k_3} - k_1$, right? Now it would be interesting to know at least $k_2$ and $k_3$, because that would show how strongly the total deformation impacts damage, compared to eq10, where the impact is linear but later in the full equations exponential, as argued in the discussion section 5.2**

These values are provided in the Appendix.

**eq10: wouldn't i make sense to treat divergence and shear separately, ie. have two different coefficients: $f_A = 1 - a_1\varepsilon_{div} - a_2\varepsilon_{shear}$, or even differential between divergence and convergence. It is clear the divergence will create open water directly, but convergence will do this to a much smaller degree (e.g. lateral divergence in convergence), and also shear should have a different coefficient.**

We assume that all deformation events (convergence, divergence and shear) indicate presence of weaker ice that may continue to be deformed. Ice weakness is simulated in neXtSIM by decreased concentration or increased damage.

Observation of any of deformation components (including convergence) is interpreted in the assimilation procedure as an increase in ice weakness and, therefore, decrease in concentration or increase in damage. We cannot find reasoning why weakening of ice due to convergence is different from weakening due to divergence or shear and, therefore, suggest that total deformation is a good proxy for detection of weak ice and a single dependence of $A$ (and $d$) on total deformation can be used. Corresponding explanations are added to the text.

**page 7 l148: simple least-squares nudging approach? Where are the least squares? Deriving eq11 from a least-square formation is possible but a little vain. I am not sure if I would call eq11 "nudging", as nudging usually implies a time varying equation such as $dv/dt = rhs + nudginpgarameter * (v_o - v)$, which is not what eq11 implies. If in a DA cycle $v_m(n)$ is computed at time t(n), then updated according to eq11, and then $v_a(n)$ is used to initialise the next DA cycle, then this is not nudging, but re-initialisation with a very simplified updating scheme. I am not criticising that, but I think that the description needs to be accurate. See also main comments.**

Rewritten as :

"We update the damage and concentration variables in the model according to the observed deformation using a simple "direct data insertion" approach as a proof of concept for DA (Stanev and Schulz-Stellenfleth, 2014)."

**l159: "the very small spatial correlation approximation is reasonable." I disagree. This assumption is valid normal to the fracture, but along the fracture, considering the nearly instantaneous fracture propagation (in nextsim and in observations), this not a "reasonable" assumption. Please rephrase.**

Rephrased as:

"Since sea ice deformation is accommodated along nearly 1D geometrical features (i.e. fractures), correlation can only usually be seen along the fracture, and so assumption of low spatial correlations in all other directions is reasonable."

**l164: value of eps_min? Mention here, that this is part of the sensitivity analysis?**

Rewritten as:

"where $\varepsilon_{min}$ is a threshold for total deformation found in sensitivity experiments."

**l171:"Since it was difficult to distinguish between the individual impacts of w_v and W in Eq. 12,", unclear, why.**

The number of experiments was increased, and values of 0, 0.5 and 1 were tested. The sentence was removed.

**l172: "0 and 1 were tested for w_d and w_A", but this means that there is no weighted average at all and all that is done is re-initialisation. I think it would help the reader to clarify the scheme: Either, there is pure re-initialisation with a somehow derived value, or no re-initialisation. The entire description of least-square nudging is entirely misleading (and does not describe, what is actually done). See main comments**

The number of experiments was increased, and values of 0, 0.5 and 1 were tested. The sentence was removed.

**page 8, l184: "difference in 90th percentile", what is that? Please be more specific.**

The 90th percentile is not used anymore for estimating predictability as it shows very similar results to $A_{MCC}$. In the sensitivity tests it is replaced with the Kolmogorov-Smirnov test applied to PDFs of forecasted and observed deformation.

**l191, related to l184. From the explanation is it no clear what is computed here, and in which sense this is different from "MCC". Is MCC a standard statistical method, or something that is only described in Korosov and Rampal, 2017? If it is a standard method, please cite a standard reference/textbook.**

Explanation of the MCC computation is added to the Appendix. A reference to a relevant textbook (Brunelli et al., 2009) is added.

**Further, there were a few metrics suggested in the cited papers by Bouchat et al 2022, and companion paper Hutter et al 2022, also Mohammadi-Aragh et al 2018. In what sense are the metrics used here related, or do they quantify entirely different properties?**

MCC is slightly different to the LKF metrics used in Hutter et al 2022 and Mohammadi-Aragh et al 2018. The following explanations are added:

"Unlike the LKF evaluation metrics suggested in (Hutter et al, 2020) that compare only statistical properties of LKFs (number, density, length, orientation, etc), the MCC-based metric estimates co-alignment of individual LKFs on model simulations and satellite observations. It is also thought to be more sensitive to LKFs with low deformation magnitude, as no threshold is applied for their detection."

**l195: "22nd January 2021" depending on the definition of "free run", I would expect that after 22 days of integration the "free run" has already quite deviated from the observations. What is the point of this comparison? Showing that the model can be "kept on track" even with simple methods, compared to not doing anything? Normally in DA, one defines a baseline/ctrl with some existing system (not the free run!) and compares how the details of the algorithm affect the solution, as has been done in section 4.2. It would also be interesting how important the observations on the current day are for forecasts, i.e. comparing a run with DA until t(n-1) to a run with DA until t(n) (where, in fact, the observations are not from one day into the future as is the case here). See also main points**

As explained above the "free run" has never seen observations of deformation and can be considered as a control simulation. The free run is not expected to match with observations (even if spatial patterns of deformation are remarkably similar). The goal of the comparison is to show that the suggested assimilation puts the model "on track", i.e. the forecasts with DA start to match with observations much better than the free run (without DA or with assimilation of something other than deformation). These aspects are also covered in replies to the major points.

**Highlight, page 9, l209: "due to its rheology" − > since only nextsim is used with one rheology, this (part of the) statement is not supported by the experiment and should be removed.**

Relation to rheology is removed.

**Also, since there is no observational data to check the results in the "unobserved" regions, one cannot claim that "neXtSIM is able to extrapolate and create realistic connections". The model simulation creates connections that look realistic, in the sense that they are not garbage, but that's about it. For a statement like the one in ll209/210, one needs experiments where part of the data is withheld from the DA, to be used later for model validation.**

As explained above, in the new set of experiments we use independent data for testing and can confirm that MCC increases also outside the area of assimilation. Nevertheless, due to the lack of sufficient cases of confirmed good extrapolation, the statement on correction of LKFs in the entire basin is rephrased as follows:

"Our experiments illustrate that even if data insertion is spatially limited by satellite observations (or even very localized in high deformation zones) it can realistically extrapolate the deformation pattern by connecting the elements of linear kinematic features."

**Further, in DA we expect that the results improve with additional data. Any other result would be failure of the DA, so all that figure 3C shows, is that the DA algorithm does, what is has been designed to do. This comparison is even further biased, because now (according to the description in Section 3, Fig2) the model has been corrected with data from the future (t(n)+24h), and then is compared to the same data from the future. I wouldn't call that prediction, but analysis.**

As explained above, in the revised manuscript observations from t(n-1)-t(n) are used for analysis on t(n) and the forecast on t(n)-t(n+1) is evaluated on independent data from the same period. The goal of Fig. 3 is exactly to illustrate that the DA algorithm does what it is supposed to do.

**page 11, Figure 4, please add colorbars to make it easier to view the images**

The colorbars are added.

**page 12, l218: persistent or persistency**

The presidency forecast is not used anymore in evaluations according to reviewers requests.

**l226: fix D_P90**

Corrected.

**l227: assimilation, better : re-initialisation.**

Rewritten as "assimilation using direct insertion"

**l233: sufficient − > a sufficient**

Corrected.

**l235: consequent −> subsequent**

Corrected.

**l238: nudging −> re-initialisation**

The entire section is rewritten and 'nudging' is not used anymore.

**page 13, l242: "The experiments with w_d cannot detect" anything, rewrite as "In the experiment with w_d, one cannot detect ..."**

Rewritten as suggested.

**Figure 6 tells me, that the leading order effect is achieved by "a_1", (except of a_1=-2, where a similar effect is achieved by modifying eps_min), so the linear relationship between total deformation and ice concentration. All other parameter appear to have small effects only. Maybe this should be stated somewhere explicitly.**

This observation is added to Section 4.3

**Fig6 is difficult to read, maybe make the bars broader?**

Figure 6 is re-plotted with results from more experiments.

**In Tab1 a_1 parameters are all positive, here, they have a negative axis, please correct, also there's seem to be experiments with a_1¡0 (i.e. to the right of 0), which are not listed in Tab1**

The sign of a1 on Figure 6 was incorrect and is corrected in the revised manuscript.

**l246: "first successful attempt", What is meant by "successful" here? This sounds like a conclusion that needs be backed with evidence. Also since the observations assimilated appear to come "from the future", the results for the first day (which is most "successful") cannot be used.**

The observations from future are not used anymore. Fig.5 evidences the success. The sentences is rewritten as follows:

"We present the first successful attempt to use observed sea ice deformation to increase accuracy of deformation prediction for the first 2 − 3 days."

**l250: "it" −> what is "it"? The relationship between deformation fields and model state variables is not shown by the DA, but by a prior correlation analysis, which I cannot evaluate, because it is moved to an appendix/supplemental material that is not accessible at the moment. Also the damage assimilation had little effect, so that questions both the empirical relation in eq9 and/or the "success" of the DA.**

Appendix is added and the sentence is rewritten as follows:

"Our study demonstrates in practice that information contained in the observed deformation fields can be used for initialisation of model state variables, and shows the time scales over which the forecast of deformation can be improved."

**l252: "proves" −> this is clearly too strong.**

Rephrased as follows:

"Our experiments illustrate that even if data insertion is spatially limited by satellite observations (or even very localized in high deformation zones) it can realistically extrapolate the deformation pattern by connecting the elements of linear kinematic features."

**l253 "corrects" −> to correct means to make it right, but there's no proof for that in the manuscript. All that the experiments show is that the model takes the initialisation information and propagates it sensibly (according to the model dynamics) into areas that have not been re-initialised. This does not mean that we now have "correct" forecasts, just that there's some "dynamical extrapolation" that needs to be evaluated with independent data (and this important step it is missing). See also main points.**

Please see previous reply.

**page 14, l266: this paragraph sounds like a project proposal with some selling arguments. Not sure if a scientific publication is the right place to advertise one's work in such a way. In my view, a scientific publication in TC should report scientific advances, but not the suitability of a system for tasks that have not yet been performed. Please rewrite or remove.**

The paragraph on practical usefulness is removed.

**page 16, l302: skill for 2–5 days − > see earlier comments, I think that the first day cannot be counted because of the data from the future.**

As seen from Fig. 6, without using data from future the accuracy of forecasts is improved for 2 - 3 days.

**l307 to the end of the section: I think that this list of factors impacting the predictive skill of LKFs would be much better placed (slightly modified) in the introduction, to lay out the scope of the manuscript and which of these aspects is addressed in the manuscript.**

The list of factors impacting predictability is added to the introduction, but a more detailed description remains in the Discussion section as it leaves many open questions for future research.

**l325: Bouillon et al., 2009 − > wrong reference. The correct Bouillon paper is from 2013, where this is called "revised EVP", although I believe that the proper reference would be Lemieux et al 2012, who were the first to modify EVP which then was described as modified EVP in Kimmritz et al (2015). It is not clear to me, how using a VP rheology (mEVP is a method to solve the VP rheology equations), that has been marked as too slow, etc. in this paper and many other papers of this group, is going to help here at all.**

The reference is corrected. The following clarifications are added:

"We expect that the model equipped with the mEVP rheology will not be capable of spatial extrapolation of the assimilated ice weakness (lowered $A$ or enhanced $d$), and that further tuning of the BBM rheology can improve the practical predictability of LKFs."

**page 17, l349: "neXtSIM is capable of extrapolating the spatially discontinuous satellite observations of deformation by connecting the elements of linear kinematic features in a realistic manner." − > this is a statement, that I think is totally justified from the evidence provided (Fig3). Please rewrite previous statements about "correcting" LKFs etc accordingly.**

The statement is rewritten (please see above).

**l351: local − > locally?**

Corrected.

**page 18 l359: Data availability: TOPAZ data and other forcing data are not mentioned, no code availability.**

The Data availability section is updated correspondingly.

**Replies to Reviewer 2**

**In this paper, damage and concentration − assumed to be a function of satellite-derived total sea ice deformation − are assimilated using a simple nudging method in the Brittle-Bingham- Maxwell (BBM) model − an off-spring of the EB/MEB model − for short-term sea ice forecast of Linear Kinematic Features. Results show that assimilation of sea ice concentration improves the skill of a short-term forecast for up to five days lead-time in comparison with persistence or a free simulation. The assimilation of damage on the other hand does not improve the skill of the forecast. It is argued that that the reason for the lack of sensitivity to damage assimilation is because the healing of ice (that reduces the damage) has a timescale that is too fast ( 24 hrs) compared with the desired lead time of the forecast (O(days)).**
**The paper is well written. The organisation and depth of the paper however is lacking. The text refers to a non-existent appendix on several occasions; the sensitivity to assimilation parameters consists of four 1 or 2-line paragraphs; there are multiple unsubstantiated statements; the results section presents inconsistent results (sea ice concentration larger than one) and focuses on small scale features in order to justify broad conclusions. The authors state that neXtSIM is not ready to assimilate deformation**

yet and that the paper is a proof-of-concept. There is no problem with this approach, but the authors have pushed this paper out for review much too early. I recommend that the paper be rejected for the moment and that authors be encouraged to resubmit later, a substantially different version of the paper that addresses (not rebut) the comments below. I am recommending a "reject with encouragement to resubmit" simply to give the author time to properly address the comments.

We appreciate the reviewer's constructive comments. As requested by the reviewers, more experiments for evaluating predictability of sea ice deformation were run (please see the replies to the first reviewer for details) in the extra time provided by the editors. The manuscript is undergoing a significant revision that can be accomplished without a rejection. All the requirements and suggestions are duly addressed without a rebuttal.

**Overstatements:**

**Line 304: "The viscous-plastic (VP) rheology used in MITgcm is known to have a less realistic slower time evolution of LKFs (Hutter et al., 2018) than the BBM rheology in neXtSIM (Olason et al., 2022). As a result, the sea ice simulated by the BBM rheology has more rapid error growth (loses skill faster) due to the correctly resolved intermittent ice motion and localised ice deformation." If the intermittent motion is correctly resolved, why is the error growing faster?**

Generally speaking, the higher is the intermittency of a stochastic process, the lower is the predictability. Therefore, a model with a low intermittency will have a tendency to overestimate intrinsic predictability.

However, our case neXtSIM is probably overestimating the intermittency. Indeed, looking in details on Fig. 13 in (Bouchat et al.,2021) we can see that the spatial structure function of neXtSIM matches the RGPS observations very well, whereas the temporal one is overestimated by 3 - 5 %. The MITgcm at 4.5 km (same or similar to as used in (Mohammadi-Aragh et al., 2018)) is worse in the spatial domain and seems better in temporal domain.

**Line 44: "In a recent model intercomparison paper (Bouchat et al., 2021), only one model, neXtSIM (neXt Generation Sea Ice Model, Bouillon and Rampal, 2015a; Rampal 45 et al., 2016), proved to be capable when run at the same spatial resolution as the available observations (i.e. ∼ 10km) to simulate the observed probability distribution, spatial distribution and fractal properties of sea ice deformation." This is incorrect. One 10-km McGill run was able to reproduce the PDF of divergence and multiple 10-km models were able to reproduce temporal scaling and spatio-temporal coupling. The only place where neXtSIM did perform better compared to other 10-km models is the spatial scaling. Intermittency does not come from the brittle parameterization in the EB/MEB/BBM. All models participating in SIREX (VP, EVP, EB) show intermittency (Bouchat et al., 2021). The source of intermittency in observed deformation is still an unresolved issue. The authors must clarify what they mean by "intermittency. If the temporal scaling exponent is used to discuss the LKFs intermittency, this is incorrect. If instead the authors are referring to LKF growth rate and lifetime, this is also incorrect. In SIREX2, it is shown that no apparent link is present between the LKF growth rates, lifetimes and the temporal scaling/multi-fractal parameters. The intermittency is revealed by the quadratic nature of the structure function.**

Following (Rampal et al., 2008) we consider the exponent $\alpha$ as a measure of the degree of intermittency.

Indeed, SIREX indicate that all models have some degree of intermittency (a slope of the temporal scaling line in log-log space) but only two low resolution models (RASM-WRF-EVP and RASM-WRF-EAP) are as close to the RGPS observations of total deformation as neXtSIM (Fig. 11.a in Bouchat et al., 2021). It is interesting to note that RASM-POP-EAP, which has exactly the same setup as RASM-WRF-EAP except for using atmospheric forcing with much lower temporal resolution, has much lower deformation rates, probably indicating the role of the atmospheric forcing in intermittency.

Nevertheless, the slope of neXtSIM temporal scaling is higher than the RGPS observations by 3 - 5 % (Fig 11.c and Fig 12.c), probably indicating overestimation of intermittency by neXtSIM. The text is rewritten as follows:

"n a recent model intercomparison paper (Bouchat et al., 2021) neXtSIM results ranked among the best for simulating the observed probability distribution, spatial distribution and fractal properties of sea ice deformation, even though it operates on a low resolution grid of 10 km. Analysis of spatial and temporal scaling (Fig. 13 in (Bouchat et al., 2021)) shows that the spatial structure function of neXtSIM matches the RGPS observations very well, whereas the temporal one is overestimated by 3 − 5 %, probably indicating some overestimation of the intermittency by neXtSIM."

**Line 209: "Thus, due to its rheology, neXtSIM is able to extrapolate and create realistic connections**

between the observed and assimilated pieces of LKFs." This is not demonstrated in the manuscript. I believe any rheology that assimilates sea ice contration (A) will show LKFs in line with observations. See Major Points below.

We agree and the sentence is removed.

**Major Points:**

**On two occasions, the authors are referring to an Appendix that is not included in the paper. The appendix must be included.**

Unfortunately, the Appendix was not added due to technical reasons. It is added in the revised manuscript.

**Equation 7-10: Sea ice concentration (A) increases in convergence (until A=1) and decreases in divergence; sea ice concentration can also increase or decrease in shear. A single dependency of A on eps_tot is therefore missing events where convergence is present along LKFs. The damage (d) dependency on eps_tot is more realistic, but the authors argue that assimilation of A is useful to increase predictive skill, whereas d is not. See below for more on this topic. The single dependence of A on (eps_tot) must be justified.**

We assume that all deformation events (convergence, divergence and shear) indicate presence of weaker ice that may continue to be deformed. Ice weakness is simulated in neXtSIM by decreased concentration or increased damage. Observation of any deformation components (including convergence) is interpreted in the assimilation procedure as an increase in ice weakness and, therefore, decrease in concentration or increase in damage. We cannot find reasoning for why weakening of ice due to convergence should be different from weakening due to divergence or shear and therefore suggest that the total deformation is a good proxy for detection of weak ice and a single dependence of $A$ (and $d$) on the total deformation can be used.

It should be added, that we have two ice categories in the model: older ice, whose concentration is used in rheology and younger ice, which is formed during water freezing and is converted to older ice only after exceeding a threshold in thickness. Only the older ice is updated in the assimilation procedure and the total ice concentration remains the same.

Corresponding explanations are added in Section 3.3.

**Line 145: The value of a1 in F_A(eps_tot) is found through sensitivity experiments using the same BBM model. This is a circular argument. This functional dependence must be derived from sea ice concentration and total deformation derived from passive microwave and Sentinel. I believe the author will find that the functional dependency is not a simple linear relationship. The author must at least show this relationship from observations and acknowledge the simplicity/caveat of the approach.**

This is not a circular argument. Indeed, $a_1$ is found in optimisation of:

$$\min(\varepsilon_{i+1}^{obs} - \varepsilon_{i+1}^{sim}) \Rightarrow a_1 \tag{1}$$

where $\varepsilon_{i+1}^{sim} = f(a_1, A_i)$. Therefore $a_1$ is a function of observed total deformation and previous concentration and is not a function of $a_1$.

We must use simulations of the same model for tuning $a_1$ for two reasons: First, reliable simultaneous observations of ice concentration and deformation at scales of 1d / 10km are not available. Second, weakening of ice (by decreased $A$ or increased $d$) must be consistent with the rheological model parametrisation. For example, if observations (in case it existed) would show a higher rate of concentration decrease per deformation rate than we obtain from simulations, then the assimilation would decrease the concentration too much, and the model would predict higher deformation than was actually observed.

Corresponding explanations are added in Section 3.4.

**Fig 5a: The skill of the model is assessed using the fraction of points where the correlation between observed and simulated deformation is significant. Statistically speaking, there will always be some points that will remain significantly correlated. The statistical significance of the signal must be shown in the figure. I also see no spatial structure in the regions of high correlations which suggest that the high-correlation points are just random events.**

It is indeed difficult to illustrate the evaluation method on an example with gaps in observations. Below (Fig. 1) is an example of maximum cross correlation (MCC) maps computed for a free run and a forecast with assimilation of synthetic observations. Only statistically significant (p=0.01) values of MCC are shown. There is an obvious spatial pattern in the MCC values linked with co-alignment of LKFs on the deformation maps. This example also shows that MCC increases in some regions (e.g. in Beaufort Sea) after assimilation where co-alignment is visually better. And in some regions it remains the same (high or low), which indicates that MCC can be used for evaluation of assimilation impact. Examples with masking of insignificant MCC and a more contrast colormap are added in the manuscript.

[Figure]

Figure 1: Impact of assimilation of synthetic observations.

**Fig 5b: I would have expected that the root mean square difference of the forecast run would asymptote to the free run. The fact that it does not is suspicious. This must be explained.**

As we are more interested in improvement of LKF localisation, and the MCC method suits better these needs, and many more experiments (free runs, assimilation of synthetic and real observations) have to be evaluated; it was decided to exclude the $D_{P90}$ metric from evaluation of the predictability. In the sensitivity experiments the $D_{P90}$ metric was replaced by the Kolmogorov-Smirnov test applied to PDFs of forecasted and observed deformation. Corresponding explanations are added to the Methodology section.

**The discussion of the sensitivity of the forecast on epsilon_min and the weighting factor w_d appears in two one-line paragraph. I suggest removing them, or a more in-depth discussion should be provided.**

The section on sensitivity to assimilation parameters is extended as new results obtained from the assimilation of synthetic observations are achieved.

**Figure 6. The constant a1 is negative. This means that the sea ice concentration is larger than 1 (see Equ 10 and since eps_tot > 0). This is not physical. The results in the figure cannot be correct.**

The sign of a1 on Figure 6 was incorrect and is corrected in the revised manuscript.

**I am assuming that the sign of a1 is incorrect. If so, A = F_A(eps_tot) = 1 − a1 * eps_tot  =0.76 for a1 = -1.2 (Line 232) and eps_tot  = 0.2 (Fig 3). For A=0.76, P\* is scaled down two orders of magnitude and the ice has no strength. Any rheological model where the ice strength is set to nearly zero along a line (an observed LKF in this case) will deform along that line – this can be tested simply. The author instead argue that the brittle rheology is key to the correct simulated location of the LKFs. This is another unsubstantiated statement.**

The statement on relation to rhelology is removed.

**Results: The simulated concentration and thickness fields after assimilation should be presented. Reading from the deformation fields and the a1 constant derived from sensitivity experiments, we should see concentration of  0.8 along LKFs, something that is not accord with RGPS observations at 10km scale resolution. I suspect assimilating damage would help producing more realistic fields. The authors give reasons for why damage assimilation is not successful, but those are not convincing. See below for more details on this topic.**

The fields of concentration and damage after assimilation are added.

The RGPS observations don't provide sea ice concentration estimates, and the current PMW observations cannot be used for accurate estimation of sea ice concentration variations at 10 km. Nevertheless, we agree that such reduction in total ice concentration is not realistic at these scales. As noted above there are two ice categories in neXtSIM: old ice and

young ice, with the former being used in rheology and with the sum of both being used in thermodynamics. Reducing the older-ice concentration for increasing of ice weakness is compensated by increasing younger-ice concentration for keeping the thermodynamic balance. Thus, older-ice concentration plays a role of sub-grid parametrisation of ice weakness that is preserved on longer time scales. As opposed to sea ice damage that is working on shorter time scales. Please also see the answer below.

**It is argued that the damage does not increase predictive skill of LKFs because ice heals too rapidly ( 24 hours). In the real world, ice heals through thermodynamic processes on much longer time scale. This choice of short healing time scale must be justified. Perhaps this is the cause of the lack of predictive associated with the assimilation of the damage.**

According to the current neXtSIM parametrisation the damage increases due to healing from 0 to 1 in 15 days. This parameter was found as optimal in experiments (Rampal et al., 2019).

In the current study, in the new experiments requested by reviewers we found that, in fact, it is not the fast damage healing that reduces the efficiency of damage assimilation. We hypothesise that concentration and damage act on different time scales. This hypothesis was tested in an idealised twin-experiment: an intact initial ice field with damage=0 and concentration=1 everywhere was 'broken-up' along realistic LKFs. In one experiment, the elements in the LKFs were initiated by reducing concentration to 0.65 and in another one - by increasing damage to 1. Variation of damage and concentration in several thousand elements of broken-up and intact ice was studied (see Fig. 9).

The study shows that in case when LKFs are initiated by reduced concentration the situation is quite simple: concentration of ice in the unbroken elements is stably high, and in the broken elements it is first low and then stably increasing due to freezing (and convergence).

For damage the situation is quite different: in the initially unbroken elements the average damage remains relatively low (0.7 - 0.85), but damage variations are very large with standard deviation reaching 0.2. This happens because in some unbroken elements, that surround the initiated cracks, the internal stress exits the Mohr-Coulomb envelope and damage increases up to 1 at very short time scales (few time steps as discussed on Fig. 7 in the manuscript). Further, a cascade of damage events occurs in the neighbours of these newly broken elements. Probability of a break up (damage increase) is higher in directions of high internal stress. Thus, the information about the initiated damage is almost instantly forgotten - it is masked by many newly damaged elements.

Large scale observations of deformation at hourly frequency could probably confirm or reject the hypothesis of how damage propagates in reality, and illustrate whether or not assimilation of damage indeed leads to a more accurate deformation field **on small time scales**. However, we assimilate and validate against daily observations that show only long term memory in ice weakness expressed in reduced ice concentration.

The details on sea ice categories, the discussion of time scales at which concentration and damage act with the accompanying figures, and the maps showing impact of assimilation on damage and older-ice concentration are added to the manuscript.

**The error is shown as 1/4 sigma. This is highly unusual. Typically, one would show an error envelope equal to one sigma (four times larger than what is shown in Figure 5).**

The figure is re-plotted to show errors of several forecasts (see Fig. 6).

**Minor Points:**

**Line 23: "Under external forcing the ice deforms primarily as an elastic material." Most deformations in the pack ice are plastic and occurs along LKFs. This sentence also contradicts the next sentence.**

The sentences are rewritten:

"As a consequence, sea ice does not drift freely anymore, but instead exhibits an intermittent drift with localised deformation. First, under increasing external forcing the undamaged ice deforms primarily as an elastic material. Internal stresses gradually accumulate in the material until a failure criterion is reached, which corresponds to a limit when sea ice fractures, and then the ice starts deforming along the multiple narrow and elongated cracks, and does so until these later refreeze or when the load (winds and currents) on the ice changes."

**Line 26: "...start deforming along multiple narrow and elongated cracks formed and does so until these later refreeze". Or when the load (winds) on the ice changes. This should be added.**

Please see the previous reply.

**Line 48: :"... the exact timing and position of strong deformation zones, or LKFs, is not yet predicted precisely". The exact position of LKFs will never be located precisely because it depends on unresolved weaknesses within the ice pack. What we can hope to reproduce is the timing, the orientation of the LKFs with respect to the large-scale forcing and their statistical distribution (width, length, density, angle of fracture, etc.). This should be corrected. This is another sentence that suggests incorrectly that BBM could eventually simulate LKFs position correctly.**

When we stated the fact that errors still exist in small-scale features in our study, we did not intend to comment on BBM's ability to eventually reproduce these features correctly. The end of the sentence is rewritten as follows:

"Despite the recent advances in the sea ice modeling, the exact timing and spatial distribution (including orientation, width, length and angle of fracture) of strong deformation zones, or LKFs, is not yet predicted precisely."

**Line 93: "The observed variables vo (damage and concentration) is computed...". "Damage" is not observed. The word "is" should read "are".**

The sentence is rewritten as follows:

"The "observed" model variables damage $d_o$ and concentration $A_o$ are derived from the observed deformation $\varepsilon_o$ using the following experimental formulations:"

**Line 94: Sometimes, the Greek lowercase epsilon symbol is used and sometimes the lunate epsilon symbols is used. The author needs to choose only one form for consistency. See Line 94 and Equations 6-7 for examples. This needs to be corrected everywhere.**

The Greek lowercase symbol ($\varepsilon$) is used everywhere.

**Line 102: It is said that CNEMS has a temporal resolution of 12 hours on Line 102; and it is said in the next sentence that it is "observed nearly every day". This sounds contradictory. This should be clarified. See Line 115 as well where 24 hours is specified.**

The sentences in Section 3.1 are rewritten as follows:

"The dataset comprises gridded products derived from Sentinel-1 synthetic aperture radar (SAR) images, with 10 km spatial resolution. Ice drift is computed from pairs of images separated by approximately 24 hours and the product is delivered every 12 hours. The spatial coverage of the product is irregular - the East Siberian, Laptev and Kara seas and the polar gap (north of 87°N) are never covered, while other Arctic regions are observed at least once nearly every 24 hours."

**Line 105: "The model is forced with the European Center for Medium-Range Weather Forecasts (ECMWF)". The version of ERA should be specified: e.g. ERA5, ERA- interim, etc.**

Version is added:

"The model is forced with the latest version (Cycle 45r1) of the Integrated Forecast System European Center for Medium-Range Weather Forecasts (ECMWF) (Owens and Hewson, 2018) and the TOPAZ4 (Sakov et al., 2014) ocean forcing fields (currents, sea surface temperature, sea surface salinity)."

**Line 117: "total" should read "total deformation".**

Corrected.

**Line 203: The units of eps_tot must be given the first time it is introduced (Table 1, Line 197). At present it is only given on Line 203.**

The units are specified in Section 2.

**Line 235: eps_tot and divergence is used interchangeably; yet they are very different. One includes shear the other not.**

The term "divergence" is used only three times: on lines 117, 144 and 233 and every time a correct relation to total deformation is given.

---

## Referee Report (RR1)

Author: A. Korosov, P. Rampal, Y. Ying, E. Olason, T. Williams
Title: Towards improving short-term sea ice predictability using deformation observations

This is a second review of the paper entitled "Towards improving short-term sea ice predictability using deformation observations". Results from the paper show that data assimilation of (reconstructed) concentration and damage derived from observed total deformation improved LKFs forecast up to two days and that the model is able to fill the gaps in the Sentinel coverage (used to calculate the total deformation) with LKFs that are qualitatively reasonable. The authors implicitely assume that the ability of the model to fill the Sentinel gaps stem from the BBM rheology by suggesting that an mEVP rheology would not be able to achieve this.

The paper is much improved, and the authors have addressed satisfactorily most of the comments from the reviewers. There is still, however, some unsubstantiated statements, misleading statements and a circular argument in the approach used in the assimilation procedure in the revised manuscript. The paper attempts two things a proof of concept that (A) assimilation can improve LKFs forecast and (B) that the BBM rheology is responsible for this improved forecast including being able to fill in the gap in Sentinel coverage, but neither is done convincingly. Both goals would be welcome contributions to the community, but in its present state the manuscript is not satisfactory. The authors should choose one option and present it in a convincing manner (see points below for details). I recommend that the paper be accepted for publications after the comments below are fully addressed (i.e., not rebutted).

**Major comment**:

General: Irrespective of the goal of the paper (Option A or B), the following points must be addressed prior to publications.

1- The study is simple. The ice strength is set to nearly zero along observed lines of high deformations (Linear Kinematic Features, LKFs), and the model in turn produces large deformation along the observed LKFs where strength is set to ~zero. Not surprisingly, the model is also able to link observed LKFs when there are observational gaps along a given LKFs (this was shown in idealized experiment by Ringeisen et al 2019). The ice strength in the BBM model has an exponential dependence on ice concentration (or open water fraction) and a linear dependence on damage. Given that total deformation cannot be assimilated in a simple manner – because it is not a prognostic variable – transfer functions (operator) are developed between simulated total deformation and simulated concentration and damage. These transfer functions are then used with observed total deformation in order to derive reconstructed damage and concentration that are used in the assimilation. This approach implicitly assumes that the relationship between deformation, sea-ice concentration and damage in the model is correct. This assumed equivalence between simulated and observed total deformation is problematic. I suggest instead:
   a. Damage cannot be observed and therefore, a transfer function must be developed from model output, as done by the authors.

b. Concentration however can be calculated directly from Sentinel-derived divergence (eps_I) contrary to what is done in the paper – see for instance work by Kwok who has produced a dataset of thin ice based on RGPS divergence as an example of how this is done. The reconstructed concentrations from observed divergence should be assimilated in the model, instead of the reconstructed concentrations obtained with the model-trained operator.

2- Another choice made by the authors is to associate total deformation with a reduction in sea ice concentration irrespective of the sign of the divergence – which is clearly not realistic. Radarsat-derived divergence PDFs are nearly symmetrical around zero with an equal number of scenes with positive and negative divergence. Presumably, the authors make this assumption because the assimilation of damage (present in both divergence and convergence) does not improve the predictability of LKFs. Instead, a lengthy and unclear discussion related to timescale is included to explain why damage does not improve predictability. This may suggest instead that the functional dependence of ice strength on damage is not appropriate. See item #3 below for more on this issue.

3- The two stated reasons for why damage does not improve LKFs forecast (different timescale; linear vs exponential dependence) are not convincing. I would argue instead that this may be an indication that damage is not correctly parameterized in the EB/MEB/BBM model.

   a. **Timescale**: The important timescale for damage to consider is not the timescale for damage growth but rather the timescale for damage healing. If damage grows and propagate in the correct direction – along the line of maximum stress (Line XX) – the benefit of that should be seen one day later.

      i. The authors states that hourly Sentinel observations would allow to test this hypothesis, but they do not exist. A feasible way to test the hypothesis would be to slow down the propagation of damage in the model and see if assimilation improves LKFs forecasts.

   b. **Exponential vs linear dependence**: The fact that assimilation of sea ice concentration has a positive impact on LKFs forecast and damage has not, suggest that perhaps ice strength should also have an exponential dependence on damage.

4- Another result that suggests that Pmax may not have the proper functional dependence on damage is the fact that the authors must resort to decreasing sea ice concentration (and therefore ice strength Pmax) whether divergence or convergence is present in the observations. The reason for implementing damage in the first place in sea ice models (Girard et al., Rampal et al., Dansereau et al., etc) is to weaken the ice early when ice deforms without the need to rely on a decrease in sea ice concentration that operate on longer a timescale, i.e. for divergent or convergent flow. The fact that sea ice concentration must be reduced even when convergence is present suggest that damage does not do its job (see Line 368). The authors appear to be missing a good opportunity to suggest a different functional dependence of Pmax on damage based on results from this assimilation procedure.

5- Figure 11: The colorbar must be defined. I suspect it shows the density of points for a given damage and eps_tot. More importantly, there is a dark yellow line (highest density of points) on the top right of the figure that shows a large number of undamaged ice (log(1-d) = 0) for very large total deformation (-1 < log(eps_tot) < 0). These points are contrary to the argument presented in the paper (see Equ 5). In panel (b), these points are

omitted (the near zero log(1-d) and log(eps_tot) ~= 0 are removed). This feature must be explained, not deleted from the panel.

6- Line 563-564: "Note, that unlike Eq. 19, the optimisation is performed here in the space of observations and using the observed total deformation." This must be clarified. The optimisation is done in the space of reconstructed diagnostics (concentration and damage) from observed total deformation (see Figure 14), not "in the space of observations".

**Option A:** Proof of concept that assimilation can improve LKFs forecast

Remove all statements from the paper that imply that the simulated LKFs shown are the result of the BBM rheology: any sea-ice model (with both compressive and shear strength) where we set the ice strength to zero along a line will deform along that line.

**Option B:** BBM is responsible for improved LKF forecast and correctly filling the gaps in sentinel coverage**.**

If the authors want to pursue that route, the following should be addressed

Line 407: "We expect that the model equipped with the mEVP rheology will not be capable of spatial extrapolation of the assimilated ice weakness (lowered A or enhanced d), and that further tuning of the BBM rheology can improve the practical predictability of LKFs". This is an unsubstantiated statement, i.e., the statement of a belief rather than a statement supported by facts. This does not have its place in a scientific paper. The authors mention on line 405-406 that the mEVP is an available option in neXtSIM: "The BBM rheology can be further tuned and compared to the modified Elasto-Visco-Plastic rheology (mEVP, Bouillon et al., 2013) that is already an available option in neXtSIM (Olason et al., 2022) for estimating the impact of rheology on the sea ice predictability." I propose that the authors run the same experiment using mEVP in order to support this statement. My own "belief" is that they will find that this is not the case. My "belief" is based on results from Ringeisen et al. (2019) showing how a standard VP rheology joins random weakness in a sea ice slab together into single LKFs feature even when the weaknesses are not oriented along lines where the maximum internal stress are present.

**Minor Points**:

1- Line 6: "We show that high values of ice deformation can be interpreted as reduced ice concentration and increased ice damage - scalar variables of neXtSIM". It can be interpreted/simulated this way but this is not the case in reality and this may be hiding a defect in the formulation of the damage parameterization. See major point above.

2- Line 45-47: "In a recent model intercomparison paper (Bouchat et al., 2021) neXtSIM results ranked among the best for simulating the observed probability distribution, spatial distribution and fractal properties of sea ice deformation, even though it operates on a low resolution grid of 10 km." The spatial resolution is lower, but the effective spatial

resolution is higher given that a finite element model can resolve discontinuities at the model grid scale, contrary to a finite difference model where 7-8 grid points are needed to resolve a discontinuity.

3- Equation 5: Are the results sensitive to the choice of exponent in the Pmax equation? I.e. power 3/2 as opposed to linear.

4- Line 196: This sentence should read: "Only the older [sea] ice [concentration] is updated in the assimilation…". The word "concentration" is missing.

5- Line 234-235: "The effect of assimilation on the prediction skill is evaluated by comparison of the simulated and observed total deformation fields as it is crucial information for safe navigation, ecological and climate studies." Ecological and climate studies have much longer time scale than the 2-days improved predictability from an assimilation approach. Only safe navigation should be kept as a motivation in this sentence.

6- Line 244: The Kolmogorov-Smirnov Distance is the difference between cumulative distribution functions (CDFs) not PDFs. This should be corrected. It is however correct that the KS distance can be used to compare two distributions and assess whether two PDFs were drawn from the same underlying probability distribution.

7- Line 371: The free run was not defined.

8- Line 378: "In the second element (orange lines on Fig. 10) the initial deformation at the break-up event is larger, the concentration decreases rapidly and, as a result, the deformation on later steps reaches much higher values." Define "rapidly". A lead opening over a 6-day time scales does not seem particularly rapid.

9- Bouchat et al. reference. Update this reference from the ESSOAR to the published version.

10- Line 561-562: "However, in Eq. 21, concentration is a function of $\varepsilon_{tot}$ assuming that ice breaks and becomes weaker due to both convergence and shear." But in observations ice become weaker when it deforms whether there is a decrease of concentration (divergence) or not (convergence).

---

## Referee Report (RR2)

Title: Towards improving short-term sea ice predictability using deformation observations
Authors: Anton Korosov, Pierre Rampal, Yue Ying, Einar Ólason, and Timothy Williams

The authors have addressed most of the comments from the previous review cycle.

There are a few comments (remnant marketing sentences as referred to by the first reviewer) that must be addressed prior to publications.

1- Abstract: It should be added that similar conclusions are obtained using a viscous plastic model (mEVP) suggesting that the improvement in LKF forecast is linked changes in mechanical properties of sea ice in the yield curve with DA, rather than the exact rheological formulation used.

2- "Line 48: The sentence is rewritten as follows: … even though the dynamic equations are solved on a low-resolution (10 km) triangular mesh using the finite element method." The authors have replaced the old sentence with a new sentence that says the exact same thing. The last part of the sentence should be removed "even though… using finite element model". My point here is that the effective resolution of a FE model is higher than the nominal resolution (dx), contrary to that of a finite difference model where several grid cells are needed to resolve a discontinuity.

3- Line 394-395: "With a similar setup (i.e. resolution, advection scheme, integration method) the model based on a viscous-plastic rheology produces forecasts with a higher error". The authors are again omitting the fact that effective and nominal resolution is different for FE and FD models. The sentence should instead read "With the same nominal resolution, advection scheme and integration method, but different discretization of the governing equation (finite difference vs finite element), the model based on …"

4- Line 330-332: "Our experiments illustrate that even if data insertion is spatially limited by satellite observations (or even very localized in high deformation zones) it can realistically extrapolate the deformation pattern by connecting the elements of linear kinematic features." While not incorrect, the sentence is misleading. I suggest adding at the end of this sentence "in accord with results from simple uniaxial loading experiment using viscous-plastic rheology (Ringeisen et al 2019). The current form of the sentence leads the reader to believe that this is the results of the rheology used. This is an obvious result; deformation (fracture) in a model will link any weaknesses within an ice field that are very roughly aligned with the large-scale forcing using any rheological that has shear and compressive strength. I am suggesting a reference from a paper on which I am co-author because this is what comes first to my mind. There must be other papers in the engineering literature that makes the same statement that can used instead.

5- Line 352: "Damage can increase from 0 to 1 in just a few model steps before it eventually starts to decay due to a mechanical healing mechanism…" Thermodynamical healing instead". If not, please clarify what mechanical healing means.

Bruno Tremblay
McGill University

---

## Author Response (AR4)

**Reviewer 1.**

The manuscript has changed substantially from the first submission, partly because the authors followed most of the reviewers' suggestions, making this nearly an entirely new manuscript. Thank you for taking my comments into account. All major problems raised in the first review have been addressed in an adequate manner.

The (new) manuscript describes data assimilation experiments with a sea ice model (neXtSIM) and deformation data in a realistic pan-Arctic configuration. The data assimilation scheme is simple, albeit effective, and the experiments clearly address potential predictability and predictive skill. The main "trick" of the assimilation scheme is the design of inverse observation operators that map the deformation data (derived from drift data) to the model variables sea ice concentration and damage (damage is a parameter unique to brittle models such as neXtSIM). The construction of these operators is described in the appendix (which was formerly not visible). My impression is that the most important result is stated in line 296: "In other words, assimilation of damage has little impact on forecast error, whereas assimilation of concentration plays a big role." The remaining results are also interesting (e.g., predictive skill of 3-4 days), but since the neXtSIM code is not publicly available, it is hard to make out how these results can be generalised and are relevant to the community except in a competitive way. The construction of the inverse observation operators depends on the neXtSIM simulations with a lot of tuning of parameters and this cannot be reproduced without access to the neXtSIM simulations (or code). The manuscript ends with a list of potential confounders and lays out ideas for future research addressing these confounders.

We are grateful for the positive evaluation of our revision! In the second revision we ran new and dedicated neXtSIM experiments with the mEVP rheology in order to illustrate that the proof-of-concept can be generalized for application in both brittle and viscous-plastic rheological frameworks. Moreover, the BBM is being implemented in open-source sea ice models such as neXtSIM-DG, SI$^3$, etc. This makes the study relevant to the wide community of sea ice modelers who use either brittle, or viscous-plastic rheological frameworks.

Data and methods are described in a clear manner, the experimental design is sound and the result presentation is mostly clear (see smaller comments below). The language is clear (but could be improved, missing article, prepositions, etc, nothing that careful editing could not fix). The authors have reduced the market place tone of the manuscript, although, to my taste, this could be done even more to not distract from the science.

We have proofread the manuscript to improve grammar and we have further toned down the sentences that sounded like 'marketing', and forcused more on the scientific discussion of our results.

The author do not make the model code of neXtSIM available. I am not sure how that complies with TC rules.

Copernicus Publications encourages authors to deposit software, algorithms, model code, etc, but this is not a strict requirement.

**Main comment**

In my understanding of the MS, the main result is that "assimilation of damage has little impact on forecast error, whereas assimilation of concentration plays a big role." (L296), because this statement gives other researchers a clear idea, which direction to follow when attempting to assimilate deformation data.

We agree that L296 is an important part of the findings in our paper, that it guides future efforts in assimilating deformation to improve sea ice LKF forecasts. However, the main result we want to show is the proof-of-concept that assimilation of observed deformation can improve LKF forecast,

which hasn't been demonstrated in the literature yet. The new experiments with the mEVP rheology reveal for the first time that the suggested DA-method is applicable for sea ice models based both on brittle and viscous-plastic rheologies, which makes the revised manuscript relevant to a wide community.

Connected to this, however, is the construction of the (inverse) observation operator, which could not be evaluated in the previous submission, as it is described in the appendix and the appendix was not part of the submission. I find the observation operators so elementary to this manuscript that I recommend moving their construction from the appendix to the main text.

We agree that the observational operators are an important part of our work and that they are not very complicated. However, we believe that the description of building the operators is quite technical, less related to the main scientific findings of the manuscript and is kept in the appendix.

From the appendix, in particular Figs12 and 15, it becomes clear that the inverse operators are incomplete (and the authors describe this). Naively I would expect that if the observation operation H_A and its inverse H_A' were perfect, panels B and C would be the same, but the reconstructed fields (panels C) are basically rescaled deformation fields (by construction). In the damage case (Fig12), this leads to a large underestimation of damage where deformation is small, and in the case of concentration to an overestimation of concentration minima (in LKFs) compared to the simulation data. In combination with the threshold eps_min in the weighting scheme (eq24), this means that only the large (approximately correct) damage values and the low (lower than simulated) concentration values are used in the assimilation.

In the manuscript we made the assumption that the fraction of older ice (responsible for ice weakening) decreases with increase of total deformation. A physically intuitive relationship would be that the total sea ice concentration linearly decreases with divergence (neglecting refreezing of opened leads, etc). However, in our work we don't use this relation between these variables.
In our method the inverse operator $H'$ is built on the assumption that the total deformation is related to the fraction of older ice (A), since that is the fraction that impacts the ice strength. Some error is allowed in $H'$ to depict this relationship, while our goal in assimilation is to minimise the error in forecasted deformation. Therefore, $H'$ is deducted not from observations, but from minimisation of the forecast error. Please also see revision suggested in the reply below.

Further, if one believes in the model dynamics (that have been used to generate the pairs in panels A and B to construct the observation operator), this method overly emphasises the low concentration in leads; assimilating these reconstructed fields will then lead to strong weaknesses and immediately allows more deformation (because ice strength/elasticity depends on concentration exponentially). This gives an (alternative) algorithmic explanation for the result. I think that this aspect requires a discussion in the text as it relates directly to the main result (to my mind) that the concentration proxy is more effective.

One should also include a discussion how realistic this effect is and if one could achieve this also in a different way. After all, there is concentration data available each day at a good resolution and I can imagine that with a similar "boost" or "enhancement" of the LKF structure in these data (i.e. a direct map between panel B and C in Fig15) they could already lead to a similar effect.

In the observations the total ice concentration is reduced due to divergence. However, as seen on Figure 15.A, the divergence is observed only in ~ 50% of all LKFs, whereas other LKFs are formed by convergence and/or shear. Since during the assimilation we decrease not the total sea ice concentration but the fraction of older ice (as one of the ice strength components), and since we want to reflect a reduction of ice strength in all LKFs, we instead use the total deformation as the predictor. Our experiments show that accounting for concentration reduction due to divergence alone decreases the effect of assimilation by 30 ~ 40 %.

The following clarifications were added to the manuscript to address these comments:

"Since reliable simultaneous observations of concentration and deformation at scales of 1 day / 10 km are not available, and damage is not an observable variable, we cannot use an empirical inverse operator. I.e., $H' \neq H'_o$ , where $H'$ is the inverse operator in question, and $H'_o$ is the empirical inverse operator between the observed total sea ice concentration ($x^o$) and sea ice deformation ($y^o$): $x^o = H'_o(y^o)$ . Ultimately, the purpose of the inverse operator is not to describe a physically realistic process (i.e., linear decrease of concentration due to divergence) but to minimize the error of the deformation forecast: $E_n^F = y_n^o - y_n = y_n^o - H \circ M \circ A \circ H'(y_{n-1}^o)$ , where $H$ is the forward operator to compute deformation from ice drift, $M$ is the numerical model to forecast ice drift, $A$ is the assimilation of the inverse operator applied to the observed total deformation at the previous time step."

On lines 190 – 195 it was emphasized that the relationship between $e_{tot}$ and $A_o$ is a reasonable assumption for improving the LKF forecasts:
"Since $A$ and $d$ are the components of sea ice strength, and the total deformation is a good proxy for the presence of weak ice, we suggest building the inverse operator $H'$ on the assumption that $A$ and $d$ are related to the total deformation."

**Minor comments, suggestions, typos:**

page 5
eq(9), not clear if the noise is added to the initial conditions ore actually to the model operator (i.e. to the solutions after integration).

The noise is added to the model operator. The sentence is rewritten as follows:
where $\psi_t$ is a random noise added to the model operator due to uncertainties in model numerics that cause the forecasts run to differ from the truth run.

l141: "The first forecast is initiated from t0"
I guess I do not understand the difference to "forecasts initiated from truth", also there's some confusion about t0 and t1, when does this (set of) run(s) actually start?

As described on lines 120 – 131 the "truth run" is initialized on $t_0$ (1 December 2020), the model is spun up until 1 January 2021, and this data is not used. Then the "truth run" continues until 31 January 2021. The 'Forecasts initiated from truth' are initiated from the output of the "truth run" on January 1st, January 2nd, January 3rd, etc.
Notation $t_1$ is replaced with $t_i$ and the following clarifications are added to eq. 14:

$$\mathbf{x}_{t_i \to t}^T = M_{t_i \to t}(\mathbf{x}_{t_i}^{tr}) + \psi_t$$

,
where $i$ denotes days in January 2021 (e.g., January 1st, January 2nd, January 3rd, etc), ...

page 6
Eq16: not consistent with other eq, e.g. eq14

The equation is replaced with a set:

$$\mathbf{x}_{t_i}^{ar} = A(\mathbf{x}_{t_i}, \mathbf{y}_{t_i}^o; H', \mathbf{w})$$

$$\mathbf{x}_{t_i \to t}^{ar} = M_{t_i \to t}(\mathbf{x}_{t_i}^{ar}) + \psi_t$$

l161: "tn + 2" latex problem. $t_{n+2}$

Corrected.

Eq17 and l162, confusion about \epsilon^A_{\delta t} and \epsilon^A_{t_n \rightarrow t_{n+1}} . I guess they are the same?

As specified on line 138 \delta t is forecast lead time. For t_n \rightarrow t_{n+1} lead time would be equal 1. The following clarifications are added:
Then the forecast is compared with deformation computed at $t_n \to t_{n+1}$ (corresponding to lead time \delta t = 1), at $t_n \to t_{n+2}$ (\delta t = 1), etc.

page 8
Eq22 needs some reformatting ("*" should probably not appear).

Corrected.

page 9
l224: "setting the weight as 0 or 1 for damage and concentration"
Table 1 implies that values of 0, 0.5, and 1 are used.

Rewritten as follows:
The variable-dependency is tested by setting the weight $w_d$ or $w_a$ to 0, i.e., letting assimilation to update either damage, or concentration, or both to see the impact of the update.

Table1: units for eps_min and a_1?

The units for eps_min ($d^{-1}$) and a_1 (% d) are added to the table.

L229: "Over 30 experiments" how were these chosen? According to table1 there are 7*3*3*3 = 189 parameter combinations.

That was a typo in the manuscript: 64 experiments were run. All effective combinations of the parameters were tested. Some ineffective combination of parameters (e.g., $w_c = 0$ and $w_d = 0$, or $w_c = 0$ and any value of $a_1$) were not tested. The tested values of parameters are also updated in the table.

page 10
L259 "that should not shock the model."
Unclear, what this is supposed to mean, please rephrase. "Shocking the model" was not a problem introduced so far.

The sentence is removed.

page 11
Figure 2. It is not quite clear what we see here. The observations (derived from drift) in panel A,B,E area compared to simulation (I am guessing without DA) in C and F, and D and G are the analysis without further integration? I.e. directly after applying eq (22) and giving the model no time to adjust to these new initial conditions?
If that's the case, this should be better described in the text.

Yes, that's the case. The caption is rewritten:
Figure 2. A: sea ice deformation (d−1) computed from observed sea ice drift between 15 and 16 January 2022. B and E: sea ice concentration and damage reconstructed from the observed deformation using Eqs. 20 and 21. C and F: concentration and damaged simulated by neXtSIM on 16 January. D and G are the analysis (Eq. 22) without further integration.

In the "analysis" of SIC (panel D), it is quite obvious that the observations replace the model where available, but otherwise there's not impact. The effects on damage (G) are probably also small (and hardly visible) because the reconstructed damage is so "featureless" (most values between 0.9 and 1)

Yes, low values of deformation, or absence of observations (i.e., due to gaps in satellite data) do not impact the analysis.

l270: "Visually the position of cracks correspond well to the observations," that is not so obvious to me.
On Jan16, the Forecast with DA looks more similar to the Forecast without DA than to the observations which have been just inserted, maybe near the Fram Strait the DA has an effect. Most strikingly the region of large deformation in the obs from the centre of the Arctic towards Alaska/N.Canada does not show up at all.

In fact, these observations were not used in assimilation. As described on lines 160 – 163, we use the observations from $t_{n-1}$ --> $t_n$ for assimilation and from $t_n$ --> $t_{n+1}$ for verification. It is therefore not surprising that the forecast looks similar to the Forecast without DA. Nevertheless, the differences in the forecasts are clearly visible and can be attributed to the assimilation. The sentence is rewritten:
Visually the position of some cracks corresponds well to the observations, …

l276: "On the third day the improvement introduced by DA is obvious only in the central Arctic, on intersection of two large cracks crossing the entire ocean."
That appears to be completely fortuitous as the two large LKFs that from a semicircle are already visible in both forecasts (with and without DA) on the day before they appear in the observations (which do not have any direct impact anymore).

It is impossible to say if this particular improvement is completely fortuitous or not. The sentence is rewritten:
On the third day the improvement which is likely introduced by DA is obvious only in the central Arctic, on the intersection of two large cracks crossing the entire ocean.

page 12
l283: "but also outside it." According to Fig4 and 5 the impact is much smaller outside of the data coverage (the color range is different from Fig 3, column 5). This should be mentioned in the text.

The sentence is rewritten:
Visual comparison of forecasts and observations, as well as the maps of MCC increase show that the correlation has improved not only in the area covered by the assimilated observations
but also outside it, although, to a lesser degree.

page 14
l293 It does not become clear (to me) from the description how the parameters are varied. Since this is a multivariate system (4 parameters), the sensitivity of one parameter may depend on the values of the other 3. How is that handled?

All effective combinations of the parameters were tested. The spread of the metric for a given parameter value indicates dependence of the metric on other parameters. We have analyzed the scatterplots of each parameter against other parameters and the metric; however, it is impossible to show all these results in the manuscript. We, therefore, limit the presentation to the mean value of the metric (Fig 7) and scatterplot of KS vs $A_{MCC}$ metric (Fig 8), which allows us to draw a conclusion on the overall variability of the metrics and the best combinations of parameters.

Fig7+8: I don't see how the presented data allows the conclusion that "w_c=1 provides the best restyle when a_1 is 0.9 or 1.2 (l312). For that you'd need a (scatter) plot with w_c and a_1 on the x and y axes, wouldn't you?

A circle is added to Fig. 8 in the revised manuscript to show two groups of points with the lowest KS and lowest $1-A_{MCC}$ that correspond to the values of parameters specified on lines 314, 315.

page 15
Figure 7. Drawing lines through 2 or 3 points is a bit misleading, bar plots or points with error bars would be more "honest", similar in Fig8 using a colorscale/range, even a discrete one, implies that more than just 2 or 3 values have been tested. Here I would use a legend with color labels.

Barplots are used instead of lines on Fig. 7. A note is added to the caption of Fig. 8 indicating that the discrete colorbar denotes a limited number of values.

l314: "Based on these observations the following values were chosen as the recommended ones"
I don't think that this "recommendation" is helpful for anyone but the authors. Given that we do not have access to the code of the model, no-one will be able to follow this "recommendation" for the sub-optimal DA procedure. I would rephrase this as "Based on these results, the best parameter choice for our configuration appears to be:"

The sentence is rephrased accordingly.

page 16
l318: "successful" I find this word in this context inappropriate. It's not clear what is "success" in this context. It sounds (again) like (over-)selling a product.

The sentence is rephrased as follows:
We present a proof-of-concept of assimilation of satellite-derived sea ice deformation which increases the accuracy of deformation prediction for the first 2 – 3 days.

l319: remove "relatively"

The term 'simple' cannot be used in the absolute manner. Our method is simpler than, e.g., EnKF, but is more complex than, e.g., direct insertion of observed concentration, therefore 'relatively simple' seems more appropriate here.

L322: "for the sake of confirming several hypotheses"
Not clear what these hypotheses are, and in the language of hypotheses testing this is impossible, you can only test and reject hypotheses.

The sentence is removed.

page 17
l356: exits -> exceeds

Corrected.

L357 "at very short time scales" -> "on very …"

Corrected.

l358: neighbours -> neighbourhood, neighbouring elements?

Replaced with "neighbor elements."

page 19
l386: "2–5 days" in Fig6 is it 3-4 days. Where do these numbers come from?

Corrected to "3 – 4".

L388 applications (missing "s"). What are "real world applications"? They would still use models, right? I would write: "more generally"

Corrected accordingly.

page 20
L433 now it is "2 – 3 days", why not stick with one set of numbers (3-4 days according to Fig6)

Corrected accordingly.

l433: remove "relatively"

The term 'simple' cannot be used in the absolute manner. Our method is simpler than, e.g., EnKF, but is more complex than, e.g., direct insertion of observed concentration, therefore 'relatively simple' seems more appropriate here.

l433: The entire sentence is weird. I would rewrite this as:
The simple data insertion approach may lead to inconsistent initial conditions because of spatial gaps in the data. Still the dynamics of the neXtSIM extrapolates the spatially discontinuous satellite observations of deformation …

The sentence is rewritten as follows:
The relatively simple data insertion approach may lead to inconsistent initial conditions because of spatial gaps in the data. The spatially discontinuous satellite observations of deformation are extrapolated by the model, connecting the elements of linear kinematic features in a realistic manner.

page 24
L545 "(in $\log_{10}(1-d)$ space)" figure11 axes labels say "lg(1-d)", please fix.

Corrected accordingly.

**Reviewer 2**

This is a second review of the paper entitled "Towards improving short-term sea ice predictability using deformation observations". Results from the paper show that data assimilation of (reconstructed) concentration and damage derived from observed total deformation improved LKFs forecast up to two days and that the model is able to fill the gaps in the Sentinel coverage (used to calculate the total deformation) with LKFs that are qualitatively reasonable. The authors implicitely assume that the ability of the model to fill the Sentinel gaps stem from the BBM rheology by suggesting that an mEVP rheology would not be able to achieve this.

The paper is much improved, and the authors have addressed satisfactorily most of the comments from the reviewers. There is still, however, some unsubstantiated statements, misleading statements and a circular argument in the approach used in the assimilation procedure in the revised manuscript. The paper attempts two things a proof of concept that (A) assimilation can improve LKFs forecast and (B) that the BBM rheology is responsible for this improved forecast including being able to fill in the gap in Sentinel coverage, but neither is done convincingly. Both goals would be welcome contributions to the community, but in its present state the manuscript is not satisfactory. The authors should choose one option and present it in a convincing manner (see points below for details). I recommend that the paper be accepted for publications after the comments below are fully addressed (i.e., not rebutted).

We are grateful to the reviewer for the valuable comments. We have introduced corresponding revisions that are relevant for the manuscript while staying within the scope of the study.

The reviewer found that the two major findings in this paper are not yet convincing. The comment on point (A) is very general and the reviewer hasn't provided further details or suggestions regarding this criticism. We believe that the experimental results presented in this paper constitute a good proof-of-concept that assimilation of deformation improves LKF forecasts. Although there is still room for improvement for both the model and the assimilation method we hope the reviewer will agree that the results support point A (in the literature, there are plenty of OSSE works based on models or DA-schemes that are imperfect, e.g., Fritzner et al., 2019; Zhang et al., 2018, etc.).

For point (B), we agree that attribution of the forecast improvements solely to the choice of model rheology is not supported by the experimental evidence. To address this, in the revised paper we ran new dedicated neXtSIM experiments with the mEVP rheology as suggested by the reviewer. We found that forecast improvements occur for both rheologies, which generalises our findings about the assimilation impact in point (A) for both brittle and viscous-plastic rheological frameworks. We also found differences in forecast performance when comparing the two rheologies that are described in the revised manuscript.

**Major comment**:
General: Irrespective of the goal of the paper (Option A or B), the following points must be addressed prior to publications.

1. The study is simple. The ice strength is set to nearly zero along observed lines of high deformations (Linear Kinematic Features, LKFs), and the model in turn produces large deformation along the observed LKFs where strength is set to ~zero. Not surprisingly, the model is also able to link observed LKFs when there are observational gaps along a given LKFs (this was shown in idealized experiment by Ringeisen et al 2019). The ice strength in the BBM model has an exponential dependence on ice concentration (or open water fraction) and a linear dependence on damage. Given that total deformation cannot be assimilated in a simple manner – because it is not a prognostic variable – transfer functions (operator) are developed between simulated total deformation and simulated concentration and damage. These transfer functions are then used with observed total deformation in order to derive

reconstructed damage and concentration that are used in the assimilation. This approach implicitly assumes that the relationship between deformation, sea-ice concentration and damage in the model is correct. This assumed equivalence between simulated and observed total deformation is problematic.

In the manuscript we made the assumption that the fraction of older ice (responsible for ice weakening) decreases with increase of total deformation. We are by no means implying this is the 'correct' relationship (i.e., physically realistic; physically intuitive). Apparently, a realistic relationship would be that the total sea ice concentration linearly decreases with divergence (and increases with convergence), if we neglect refreezing of opened leads, rafting of thinner ice, and other processes. However, in our work we don't use this relation between these variables. In other words, $H' \neq H'_o$ , where $H'$ is the inverse operator in question, and $H'_o$ is the empirical inverse operator between the observed total sea ice concentration ($x^o$) and sea ice divergence ($y^o$): $x^o = H'_o(y^o)$ .

In our method the inverse operator $H'$ is built on the assumption that the total deformation is related to the fraction of older ice ($A_o$), since that is the fraction that impacts the ice strength. Some error is allowed in $H'$ to depict this relationship, while our goal in assimilation is to minimise the error in forecasted deformation. Therefore, $H'$ is deducted not from observations, but from minimisation of the forecast error $E^F_n$:

$$H' = \underset{H'}{\mathrm{argmin}}(E_n^F)$$

$$E_n^F = y_n^o - y_n = y_n^o - H \circ M \circ A \circ H'(y_{n-1}^o)$$

,

where $y_n$ is forecasted total deformation at step $t=n$, $y_n^o$ is observed total deformation, $H$ is the forward operator to compute deformation from ice drift, $M$ is the numerical model to forecast ice drift and $A$ is the assimilation of the inverse operator $H'$ applied to the observed total deformation $y_{n-1}^o$ at the previous time step. As seen from the above formulae, there are no circular arguments.

In Olason et al. (2022) we showed that both the magnitude, the spatial pattern, the temporal evolution, and the spatial scaling of the simulated deformation fields have quite high accuracy when compared to the RGPS observations. This provided the basis for assimilation, since the model simulation can reproduce a lot of the important features in the observation. Although the simulated relationship is not perfect, it is accurate to a first order to allow the assimilation approach to improve forecasts.

The following clarifications were added to the manuscript to address the reviewer's comment:

Line 88: "It should be noted that there are two ice categories in the model: young ice, which is formed during water freezing, and older ice which is formed after young ice exceeds a threshold in thickness (Rampal et al., 2019). Only the older ice concentration (referred to as $A$) is used in the rheological equations."

Line 103: "The "observed" model variables damage $d_o$ and concentration of older ice $A_o$

Lines 173 – 179 were rewritten:

"Since reliable simultaneous observations of concentration and deformation at scales of 1 day / 10 km are not available, and damage is not an observable variable, we cannot use an empirical inverse operator. I.e., $H' \neq H'_o$, where $H'$ is the inverse operator in question, and $H'_o$ is the empirical inverse operator between the observed total sea ice concentration ($x^o$) and sea ice deformation ($y^o$): $x^o = H'_o(y^o)$. Ultimately, the purpose of the inverse operator is not to describe a physically realistic process (i.e., linear decrease of concentration due to divergence) but to minimize the error of the deformation forecast:
$E^F_n = y^o_n - y_n = y^o_n - H \circ M \circ A \circ H'(y^o_{n-1})$, where $H$ is the forward operator to compute deformation from ice drift, $M$ is the numerical model to forecast ice drift, $A$ is the assimilation of the inverse operator applied to the observed total deformation at the previous time step."

On lines 190 – 195 it was emphasized that the relationship between $e_{tot}$ and $A$ is a reasonable assumption for improving the LKF forecasts:
"Since $A$ and $d$ are the components of sea ice strength, and the total deformation is a good proxy for the presence of weak ice, we suggest building the inverse operator $H'$ on the assumption that $A$ and $d$ are related to the total deformation."

> I suggest instead:
> a. Damage cannot be observed and therefore, a transfer function must be developed from model output, as done by the authors.
> b. Concentration however can be calculated directly from Sentinel-derived divergence (eps_I) contrary to what is done in the paper – see for instance work by Kwok who has produced a dataset of thin ice based on RGPS divergence as an example of how this is done. The reconstructed concentrations from observed divergence should be assimilated in the model, instead of the reconstructed concentrations obtained with the model-trained operator.

The approach of Kwok et al. for computing the total sea ice concentration is based on the aforementioned 'correct' linear relation between divergence (divergence rate multiplied by forecast time) and decrease of the total sea ice concentration. However, as seen on Figure 15.A, the divergence is observed only in ~ 50% of all LKFs, whereas other LKFs are formed by convergence and/or shear. Since we decrease not the total sea ice concentration but the fraction of older ice (as one of the ice strength components), and since we want to reflect a reduction of ice strength in all LKFs, we instead use the total deformation as the predictor. Our experiments show that accounting for concentration reduction due to divergence alone decreases the effect of assimilation by 30 ~ 40 %.

> 2. Another choice made by the authors is to associate total deformation with a reduction in sea ice concentration irrespective of the sign of the divergence – which is clearly not realistic. Radarsat-derived divergence PDFs are nearly symmetrical around zero with an equal number of scenes with positive and negative divergence. Presumably, the authors make this assumption because the assimilation of damage (present in both divergence and convergence) does not improve the predictability of LKFs. Instead, a lengthy and unclear discussion related to timescale is included to explain why damage does not improve predictability. This may suggest instead that the functional dependence of ice strength on damage is not appropriate. See item #3 below for more on this issue.

In our method the inverse operator $H'$ is built on the assumption that the total deformation is related to the fraction of older ice (A), since that is the fraction that impacts the ice strength. Our experiments

show that accounting for older ice concentration reduction due to divergence alone, decreases the effect of assimilation by 30 ~ 40 %.

With a scalar damage unaffected by other sea ice processes (unlike the concentration), the dependance of the ice strength on damage is completely arbitrary, since the damage and stress must evolve in a manner that is consistent with the constitutive relation (i.e. the effective strength after damaging should be the same regardless of whether we have a factor 1-*d* or 1-*f(d)*, where *f* is any function that increases monontonically from 0 to 1 with *d*). What may be sub-optimal in our model is either the constitutive relation itself or the numerical implementation of the co-evolution of stress, damage and concentration, and we now say this in our revised discussion section. Having said this, the goal of the manuscript is to show that despite having an imperfect model and an imperfect DA-method we can still improve the LKF forecast.

3. The two stated reasons for why damage does not improve LKFs forecast (different timescale; linear vs exponential dependence) are not convincing. I would argue instead that this may be an indication that damage is not correctly parameterized in the EB/MEB/BBM model.

   a. **Timescale**: The important timescale for damage to consider is not the timescale for damage growth but rather the timescale for damage healing. If damage grows and propagate in the correct direction – along the line of maximum stress (Line XX) – the benefit of that should be seen one day later.

      i. The authors states that hourly Sentinel observations would allow to test this hypothesis, but they do not exist. A feasible way to test the hypothesis would be to slow down the propagation of damage in the model and see if assimilation improves LKFs forecasts.

   b. **Exponential vs linear dependence**: The fact that assimilation of sea ice concentration has a positive impact on LKFs forecast and damage has not, suggest that perhaps ice strength should also have an exponential dependence on damage.

We agree with these comments and thank the reviewer for sharing his thoughts. Slowing down the damage propagation or changing the dependence of ice strength on damage are feasible ways forward that may potentially improve the effect of assimilation. However, testing of these approaches goes outside the scope of the manuscript, which aim is to provide a proof-of-concept for improving prediction of LKFs by assimilation of satellite-derived deformations using an existing state-of-the-art sea ice model and the method presented in this manuscript. We added to the list of possible future steps in the Discussion section.

4. Another result that suggests that Pmax may not have the proper functional dependence on damage is the fact that the authors must resort to decreasing sea ice concentration (and therefore ice strength Pmax) whether divergence or convergence is present in the observations. The reason for implementing damage in the first place in sea ice models (Girard et al., Rampal et al., Dansereau et al., etc) is to weaken the ice early when ice deforms without the need to rely on a decrease in sea ice concentration that operate on longer a timescale, i.e. for divergent or convergent flow. The fact that sea ice concentration must be reduced even when convergence is present suggest that damage does not do its job (see Line 368). The authors appear to be missing a good opportunity to suggest a different functional dependence of Pmax on damage based on results from this assimilation procedure.

We thank the reviewer for this comment. We agree with the reviewer that the functional dependence of Pmax on damage can potentially be improved and that our study allowed the potential impact of this improvement to be identified. However, we think that addressing this properly would require a significant and therefore dedicated study. We thus decided to add to the discussion section this interesting point as a potential for future studies.

5. Figure 11: The colorbar must be defined. I suspect it shows the density of points for a given damage and eps_tot. More importantly, there is a dark yellow line (highest density of points) on the top right of the figure that shows a large number of undamaged ice (log(1-d) = 0) for very large total deformation (-1 < log(eps_tot) < 0). These points are contrary to the argument presented in the paper (see Equ 5). In panel (b), these points are omitted (the near zero log(1-d) and log(eps_tot) ~= 0 are removed). This feature must be explained, not deleted from the panel.

We thank the reviewer for spotting this. There were indeed a few elements that were erroneously selected for training the damage-deformation relation. After fixing a bug in the selection code these elements disappeared without changing the results of the fitting. A colorbar was added to the figure.

6. Line 563-564: "Note, that unlike Eq. 19, the optimisation is performed here in the space of observations and using the observed total deformation." This must be clarified. The optimisation is done in the space of reconstructed diagnostics (concentration and damage) from observed total deformation (see Figure 14), not "in the space of observations".

The sentence is rewritten: "... the optimization is performed here in the space of observed and

simulated total deformation: $\max\limits_{H'}(A_{MCC}(\varepsilon_{tot}, \varepsilon_{tot}^o))$ .

**Option A:** Proof of concept that assimilation can improve LKFs forecast

Remove all statements from the paper that imply that the simulated LKFs shown are the result of the BBM rheology: any sea-ice model (with both compressive and shear strength) where we set the ice strength to zero along a line will deform along that line.

**Option B:** BBM is responsible for improved LKF forecast and correctly filling the gaps in sentinel coverage.

If the authors want to pursue that route, the following should be addressed

Line 407: "We expect that the model equipped with the mEVP rheology will not be capable of spatial extrapolation of the assimilated ice weakness (lowered A or enhanced d), and that further tuning of the BBM rheology can improve the practical predictability of LKFs". This is an unsubstantiated statement, i.e., the statement of a belief rather than a statement supported by facts. This does not have its place in a scientific paper. The authors mention on line 405-406 that the mEVP is an available option in neXtSIM: "The BBM rheology can be further tuned and compared to the modified Elasto-Visco-Plastic rheology (mEVP, Bouillon et al., 2013) that is already an available option in neXtSIM (Olason et al., 2022) for estimating the impact of rheology on the sea ice predictability." I propose that the authors run the same experiment using mEVP in order to support this statement. My own "belief" is that they will find that this is not the case. My "belief" is based on results from Ringeisen et al. (2019) showing how a standard VP rheology joins random weakness in a sea ice slab together into single LKFs feature even when the weaknesses are not oriented along lines where the maximum internal stress are present.

As suggested by the reviewer we ran assimilation experiments with the mEVP rheology in place of the BBM, using the same numerical implementation and parameters' values as in Olason et al. 2022. In the experiments we assimilate deformation only through reduction of sea ice concentration. Two experiments were run: one with the same values of parameters of the assimilation method as in the "best" experiment with BBM, and another experiment with a higher value of $a_1$. Experiments show (see Figure 11 in the revised manuscript) that with the same assimilation setup the error of the EVP-based forecast for the first day is the same as for the BBM-based forecast. However, for larger lead

time the error grows faster and saturates at a higher level in the mEVP simulation (see Figure 12 in the revised manuscript).

In light of these new results, we thus think that our manuscript now could be **Option C**: it is a proof of concept of a generalized approach that observed deformation can be assimilated to improve forecast of sea ice deformation in both brittle and viscous-plastic sea ice models. Lines 405 - 409 are rewritten as follows:

"Tuning of the BBM rheology regarding the speed of damage propagation and changing the dependence of ice strength or Pmax on damage could improve the practical predictability of LKFs. The mEVP rheology could also be further tuned by changing the number of sub-cycles or adding a damage parameter (as in Savard et al., 2023)."

A sub-section "Experiments with mEVP rheology" was also added to the Discussions section in order to describe these new findings.

**Minor Points**:

Line 6: "We show that high values of ice deformation can be interpreted as reduced ice concentration and increased ice damage - scalar variables of neXtSIM". It can be interpreted/simulated this way but this is not the case in reality and this may be hiding a defect in the formulation of the damage parameterization. See major point above.

We agree that there is room for improvement of the damage parametrization. However, that does not prevent the assimilation of deformation through modifying either or both concentration and damage as the two components of sea ice strength. Given that reduction of concentration can be used in a more generalized approach in both EB and EVP rheologies the sentence is rewritten as follows:

"A new method for assimilation of satellite-derived sea ice deformation into numerical sea ice models is presented. … We show that high values of ice deformation can be interpreted as reduced ice concentration or increased ice damage - scalar variables responsible for ice strength in brittle or viscous-plastic sea ice dynamical models. This method is tested as a proof-of-concept with the neXt-generation Sea Ice Model (neXtSIM), where the assimilation scheme uses a data insertion approach and forecasting with one member."

Line 45-47: "In a recent model intercomparison paper (Bouchat et al., 2021) neXtSIM results ranked among the best for simulating the observed probability distribution, spatial distribution and fractal properties of sea ice deformation, even though it operates on a low resolution grid of 10 km." The spatial resolution is lower, but the effective spatial resolution is higher given that a finite element model can resolve discontinuities at the model grid scale, contrary to a finite difference model where 7-8 grid points are needed to resolve a discontinuity.

The sentence is rewritten as follows: "… even though the dynamic equations are solved on a low-resolution (10 km) triangular mesh using the finite element method."

Equation 5: Are the results sensitive to the choice of exponent in the Pmax equation? I.e. power 3/2 as opposed to linear.

The power 3/2 is for thickness in Eq. 5, it has no impact on the results where concentration and damage are modified.

Line 196: This sentence should read: "Only the older [sea] ice [concentration] is updated in the assimilation...". The word "concentration" is missing.

The sentence is updated accordingly.

Line 234-235: "The effect of assimilation on the prediction skill is evaluated by comparison of the simulated and observed total deformation fields as it is crucial information for safe navigation, ecological and climate studies." Ecological and climate studies have much longer time scale than the 2-days improved predictability from an assimilation approach. Only safe navigation should be kept as a motivation in this sentence.

The sentence is updated accordingly.

Line 244: The Kolmogorov-Smirnov Distance is the difference between cumulative distribution functions (CDFs) not PDFs. This should be corrected. It is however correct that the KS distance can be used to compare two distributions and assess whether two PDFs were drawn from the same underlying probability distribution.

The sentence is updated accordingly.

Line 371: The free run was not defined.

"Free run" is replaced by "truth run ($x^{tr}_t$)"

Line 378: "In the second element (orange lines on Fig. 10) the initial deformation at the break-up event is larger, the concentration decreases rapidly and, as a result, the deformation on later steps reaches much higher values." Define "rapidly". A lead opening over a 6-day time scales does not seem particularly rapid.

The scale of the X-axis is given in format MM-DD HH and concentration decreases by almost 10% in 6 hours. The sentence is rewritten: "… the concentration decreases rapidly (at a rate of ~ 1%/hour) …" and the note on axis format is added to the figure caption: "The scale of the X-axis is given in format MM-DD HH."

Bouchat et al. reference. Update this reference from the ESSOAR to the published version.

The reference was updated.

Line 561-562: "However, in Eq. 21, concentration is a function of $\varepsilon_{tot}$ assuming that ice breaks and becomes weaker due to both convergence and shear." But in observations ice become weaker when it deforms whether there is a decrease of concentration (divergence) or not (convergence).

The sentence is rewritten as follows:

However, in Eq. 21, the concentration of older sea ice is a function of $\varepsilon_{tot}$ assuming that ice breaks and becomes weaker due to both convergence, divergence and shear.

---

## Author Response (AR5)

**Replies to Reviewer 1**

The authors have addressed most of the comments from the previous review cycle. There are a few comments (remnant marketing sentences as referred to by the first reviewer) that must be addressed prior to publications.

All minor comments provided by the reviewer are addressed.

**Abstract: It should be added that similar conclusions are obtained using a viscous plastic model (mEVP) suggesting that the improvement in LKF forecast is linked changes in mechanical properties of sea ice in the yield curve with DA, rather than the exact rheological formulation used.**

The following sentences are added to the Abstract:

Similar conclusions are obtained using both brittle and viscous-plastic rheologies implemented in neXtSIM. Thus, the forecasts improve due to the update of sea ice mechanical properties, rather than the exact rheological formulation.

**Line 48: The sentence is rewritten as follows: ... even though the dynamic equations are solved on a low-resolution (10 km) triangular mesh using the finite element method." The authors have replaced the old sentence with a new sentence that says the exact same thing. The last part of the sentence should be removed "even though... using finite element model". My point here is that the effective resolution of a FE model is higher than the nominal resolution (dx), contrary to that of a finite difference model where several grid cells are needed to resolve a discontinuity.**

The second part of the sentence "..., even though the dynamic equations are solved on a low-resolution (10 km) triangular mesh using the finite element method" was replaced with a sentence that contains only factual information: "In the aforementioned studies neXtSIM was run with a similar setup: the dynamic equations were solved on at triangular mesh with 10 km resolution using finite element method."

**Line 394-395: "With a similar setup (i.e. resolution, advection scheme, integration method) the model based on a viscous-plastic rheology produces forecasts with a higher error". The authors are again omitting the fact that effective and nominal resolution is different for FE and FD models. The sentence should instead read "With the same nominal resolution, advection scheme and integration method, but different discretization of the governing equation (finite difference vs finite element), the model based on ..."**

The reviewer's statement is not correct here. We run both brittle and viscous-plastic model using Lagrangian advection scheme with FE discretisation. Therefore not only the nominal, but also the effective resolutions are the same. The only thing which is different is the rheological formulation. The following clarifications are added to the sentence:

With a similar setup (i.e. spatial resolution equals 10 km, Lagrangian advection scheme, Finite Element integration method) the model based on a viscous-plastic rheology produces forecasts with a higher error.

**Line 330-332: "Our experiments illustrate that even if data insertion is spatially limited by satellite observations (or even very localized in high deformation zones) it can realistically extrapolate the deformation pattern by connecting the elements of linear kinematic features." While not incorrect, the sentence is misleading. I suggest adding at the end of this sentence "in accord with results from simple uniaxial loading experiment using viscous-plastic rheology (Ringeisen et al 2019). The current form of the sentence leads the reader to believe that this is the results of the rheology used. This is an obvious result; deformation (fracture) in a model will link any weaknesses within an ice field that are very roughly aligned with the large-scale forcing using any rheological that has shear and compressive strength. I am suggesting a reference from a paper on which I am co-author because this is what comes first to my mind. There must be other papers in the engineering literature that makes the same statement that can used instead.**

The following text was added:

..., in accord with results from a simple uniaxial loading experiment using viscous-plastic rheology (Ringeisen et al 2019).

**Line 352: "Damage can increase from 0 to 1 in just a few model steps before it eventually starts to decay due to a mechanical healing mechanism..." Thermodynamical healing instead". If not, please clarify what mechanical healing means.**

"mechanical" is replaced with "thermodynamical".